# LEARNING FLOW-GUIDED REGISTRATION FOR RGB–EVENT SEMANTIC SEGMENTATION

## ABSTRACT

Event cameras capture microsecond-level motion cues that complement RGB sensors. However, the prevailing paradigm of treating RGB-Event perception as a fusion problem is ill-posed, as it ignores the intrinsic (i) *Spatiotemporal* and (ii) *Modal Misalignment*, unlike other RGB-X sensing domains. To tackle these limitations, we recast RGB-Event segmentation from fusion to registration. We propose BRENet, a novel flow-guided bidirectional framework that adaptively matches correspondence between the asymmetric modalities. Specifically, it leverages temporally aligned optical flows as a coarse-grained guide, along with fine-grained event temporal features, to generate precise forward and backward pixel pairings for registration. This pairing mechanism converts the inherent motion lag into terms governed by flow estimation error, bridging modality gaps. Moreover, we introduce Motion-Enhanced Event Tensor (MET), a new representation that transforms sparse event streams into a dense, temporally coherent form. Extensive experiments on four large-scale datasets validate our approach, establishing flow-guided registration as a promising direction for RGB-Event segmentation.

## 1 INTRODUCTION

Semantic segmentation, the task of assigning pixel-wise semantic categories, has received much research attention. It is fundamental in computer vision tasks (Wu et al., 2025; 2024), e.g., medical imaging (Chen et al., 2022; Ginley et al., 2023) and robotics (Mosbach & Behnke, 2024; Panda et al., 2023). While most approaches focused on RGB modality, recent researchers have explored incorporating event cameras (Liang et al., 2023), e.g., Dynamic Vision Sensor (DVS). Event cameras are bio-inspired devices that asynchronously capture edge motion with higher temporal resolution (10 µs vs 3 ms), higher dynamic range (120 dB vs 60 dB), and lower latency (Shiba et al., 2022). These inherent advantages enable robust motion estimation under challenging real-world scenarios.

Despite advances in multimodal integration modules, prior methods (Zihao Zhu et al., 2018; Yao & Chuah, 2024; Chen et al., 2023) treat RGB-Event perception as a fusion problem that implicitly assumes spatiotemporal co-registration. However, RGB-Event perception is intrinsically unaligned, unlike other RGB-X sensing domains: (1) *Spatiotemporal Misalignment*: Events are captured with microsecond-level latency while RGB images are sampled at a lower sampling rate, causing spatial shifts of corresponding scene points. (2) *Modal Misalignment*: Events record asynchronous, sparse brightness changes (temporal derivatives) while RGB captures synchronous, dense absolute intensities. This sensing mismatch presents significant challenges for pixel-level segmentation tasks that require spatially coherent, dense visual information. As illustrated in Figure 1, fusion-centric approaches overlook that RGB and event streams are intrinsically misaligned: fusing multiple event frames with an RGB image disrupts motion continuity and leaves modality gaps unresolved. Without explicit registration, these fusion-centric pipelines yield suboptimal predictions, limiting segmentation performance. Visualizations of misalignments are in the Appendix.

To address these misalignments, we recast RGB-Event perception as a registration problem rather than fusion, following a "*registers first, then fuses*" design principle. We propose BRENet, a registration-centric **B**idirectional **R**GB-**E**vent Semantic Segmentation framework that estimates correspondence fields in forward and backward flows to register asynchronous event streams to the reference RGB frame. A subsequent Temporal Fusion Module (TFM) performs spatial warping to correct spatiotemporal offsets and locate motion occlusions. The flow-guided bidirectional registration mechanism provides ensemble temporal cues and yields temporally coherent features, substantially

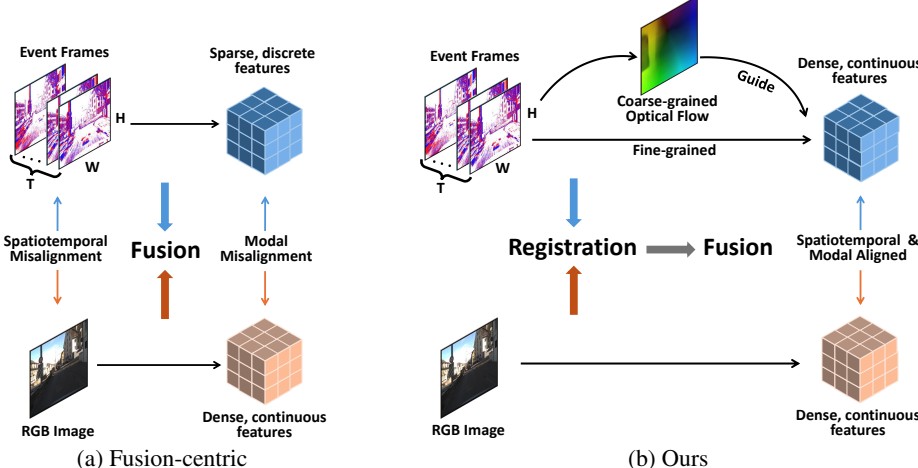

Figure 1: **Comparisons between existing fusion-centric models and our registration-centric method.** (a) Fusion-centric methods assume spatiotemporal co-registration and ignore inherent misalignments. (b) In contrast, we rethink RGB-Event segmentation with *registers first, then fuses* design principle, mitigating both misalignments.

mitigating the ***Spatiotemporal Misalignment***. Unlike fusion-centric approaches, registration is not treated as an auxiliary enhancement but as the core objective of the architecture.

Optical flow serves as a coarse-grained motion prior that guides registration and offers three benefits: (i) it reconciles sampling-rate disparity through temporal alignment; (ii) it converts sparse, derivative-like events into dense representations that are compatible with RGB; and (iii) it captures motion changes that align with the inherent modal nature of events. Building on this, we introduce Motion-Enhanced Event Tensors (MET). This novel representation enhances coarse global motion from optical flows and fine temporal event cues, performing representation-level registration (RGB and event). It alleviates the ***Modal Misalignment*** while preserving low-level details in a multi-granularity manner, alleviating both misalignments.

In summary, our contributions in this paper include:

- We formulate RGB-Event semantic segmentation as a *registration* problem. We propose a novel flow-guided registration-centric framework, **BRENet**, which estimates pixel-wise bidirectional correspondences to pair the asynchronous event stream with the reference RGB frame. It then leverages a Temporal Fusion Module (TFM) for adaptive fusion. Rather than fusing RGB-Event data directly, BRENet ***registers first, then fuses***, shifting the paradigm from a fusion-centric to a registration-centric approach.

- We introduce a new event representation, Motion-enhanced Event Tensor (MET), to integrate coarse-grained optical flows with fine-grained temporal visual cues. We redefine the role of optical flow: not as an alternative input modality, but as a bridge that dynamically aligns events with RGB. To the best of our knowledge, we are the first to employ optical flows for registration in RGB-Event perception.

- We evaluate our proposed BRENet on DDD17, DSEC, DELIVER, and M3ED datasets and demonstrate its effectiveness. Compared to SOTA models, BRENet achieves superior performance.

## 2 RELATED WORK

### 2.1 RGB-EVENT SEMANTIC SEGMENTATION

Event modality provides complementary information to RGB modality, offering a new perspective on motion dynamics. However, as discussed in Section 1, two key misalignments arise when integrating these two modalities, reflecting differences in sparsity, frequency, and viewpoint.

To address these misalignments, researchers develop various fusion-centric methods (Chen et al., 2021; 2024) to integrate multi-modal features. CMX (Zhang et al., 2023a) presents a Cross-modal Feature Rectification Module that uses one modality to rectify and refine multi-modal features, learning long-range contextual information. EVSNet (Yao & Chuah, 2024) learns short- and long-term temporal motions from event and then aggregates multi-modal features adaptively. EventSAM (Chen et al., 2023) presents a cross-modal adaptation model of SAM (Kirillov et al., 2023) for event modality, leveraging weighted knowledge distillation. HALSIE (Das Biswas et al., 2024) proposed a dual-encoder framework with Spiking Neural Network (SNN) and Artificial Neural Network (ANN) to improve cross-domain feature aggregation. SpikingEDN (Zhang et al., 2024) designs an efficient SNN model that employs a dual-path spiking module for spatially adaptive modulation.

Despite these advances, fusion-centric approaches assume implicit spatiotemporal co-registration and therefore ignore misaligned signals caused by temporal discontinuity and motion displacement. In contrast, our approach is registration-centric: we establish pixel-wise correspondences via flow-guided bidirectional pairing. This design ideology converts inherent ***Spatiotemporal Misalignment*** into learnable parameters of a registration module, which can be optimized in a model-agnostic way.

## 2.2 EVENT REPRESENTATION

Researchers have explored different event representations. Each single event $e_i$ is represented as a 4-tuple: $e_i = [x_i, y_i, p_i, t_i]$ where $x_i, y_i$ are spatial coordinates, $t_i$ is the timestamp and $p_i \in \{-1, +1\}$ indicates the polarity of brightness change (increasing or decreasing).

Early works introduce an image-based representation of event streams. Rebecq et al. (2017) consider events in overlapping windows and yield motion-compensated event representation. EV-FlowNet (Zhu et al., 2018) processes the event streams to event frames where the value of each pixel is the number of events. Maqueda et al. (2018) separates all events into two streams on polarity (positive and negative) and then designs a dual-branch model.

Following image-based representations, grid-based representation has been introduced. Zhu et al. (2019) discretizes event streams into bins and stacks all event bins to generate voxel grids. EST (Gehrig et al., 2019) samples voxel grids from event streams and learns the event representation using differentiable operations. Grid-based representations discretize the temporal dimension into discrete $B$ bins and accumulate all events in a fixed time interval $\Delta t$. Recent approaches explore refined representations based on voxel grids. EISNet (Xie et al., 2024a) leverages event counts as indicators of scene activity, capturing activity-aware features. SE-Adapter (Yao et al., 2024) proposes MSP, a multi-scale spatiotemporal feature-enhanced event representation. However, these representations overlook that the asynchronous and sparse nature of event is incompatible with pixel-level segmentation tasks, which require dense visual information.

In this work, we tackle these limitations by proposing MET, which converts sparse, discrete event data to dense, continuous features. It reframes optical flows as a structural prior to register event frames into the reference RGB timestamp. By integrating coarse-grained flow information with fine-grained temporal correlations, MET effectively mitigates modal gaps and generates more robust features.

# 3 METHODOLOGY

## 3.1 MOTIVATION

Existing fusion-centric methods (Xie et al., 2024a), discretize time and accumulate polarity counts, integrating RGB-Event without establishing pixel-wise correspondences. This leaves both spatiotemporal discrepancy (motion-induced spatial replacement) and the modal gap unresolved.

Assume RGB frame $I$ captures at time $t_k$, all events $E$ are continuously recorded during $[t_{k-1}, t_k]$, event location is $x$ and velocity is $v$. The fusion function can be expressed as:

$$\mathcal{F}_{\text{fuse}}\big(I_{t_k}, E_{[t_{k-1}, t_k]}\big) = \sum_i w_i \, f\Big(I_{t_k}(x_0), \, E_i(x_0 + v \cdot (t_k - t_i))\Big), \qquad t_k - t_i \geq 0. \quad (1)$$

where $w$ are the fusion weights. Assume an average motion lag $\bar{\delta}$, weighted spatial shift $\Delta_{\text{fuse}}(x)$ is:

$$\|\Delta_{\text{fuse}}(x)\| = \|\sum_i w_i \cdot v_i \cdot (t_k - t_i)\| \approx \|\sum_i w_i \cdot v_i \cdot \bar{\delta}\| > 0 \quad (2)$$

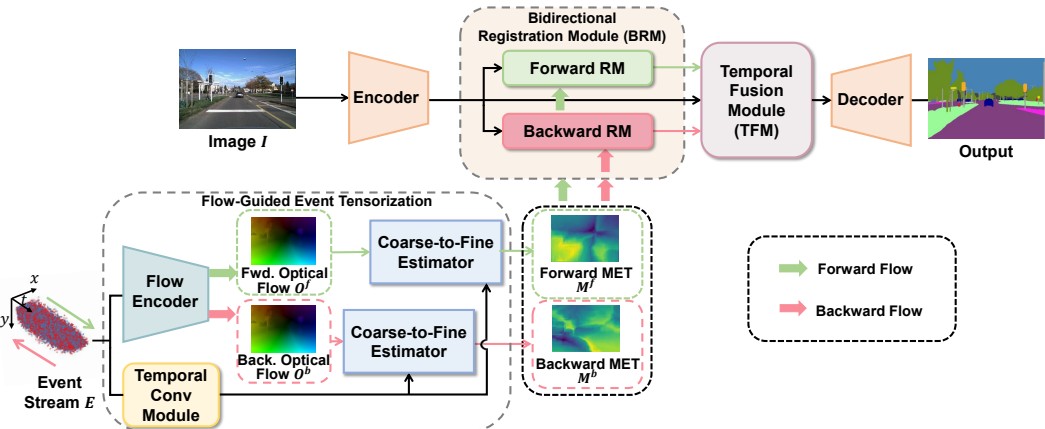

Figure 2: **Illustration of overall framework.** Given an input RGB-Event pair, the Coarse-to-Fine Estimator (CFE) generates bidirectional METs through coarse-grained optical flows and fine-grained event temporal features. The Bidirectional Registration Module (BRM) further adaptively registers METs into image features in both forward and backward directions. Finally, the Temporal Fusion Module (TFM) fuses bidirectional features to learn the temporal consistency.

Given the positive temporal lag $\bar{\delta} > 0$ and the velocities of moving objects $v_i > 0$, learnable fusion weights $w$ cannot eliminate the irreducible spatial shift between RGB and events.

Inspired by the success of optical flow in searching for shared regions in video stabilization (Yu & Ramamoorthi, 2020; Shi et al., 2022; Zhao et al., 2023), we employ it to address misalignments and provide visual guidance. In our registration-centric design, optical flow offers key advantages: (i) registering events onto the RGB frame grid with temporal alignment of the asynchronous event stream; (ii) generating dense motion fields that eliminate sparsity; (iii) transforming per-pixel intensity changes (log intensity) to modality-agnostic motion vectors in pixel space, analogous to RGB-derived features (Bardow et al., 2016). This paradigm shift is inherently suited for visual tasks with dense RGB information and can effectively mitigate both *Spatiotemporal* (i) and *Modal Misalignment* (ii & iii) by providing motion and correspondence information (Shi et al., 2023).

Specifically, we convert the irreducible spatial shift to optical flow estimation errors through flow-guided registration. Define the warping function as:

$$\phi_{t \to t_0}(x) = x - \int_t^{t_0} u(x, \tau) d\tau \tag{3}$$

where $u$ is the optical-flow field under a local constant-velocity approximation. Thus, the displacement after registration can be represented as:

$$\|\Delta_{\text{reg}}(x)\| = \left(\phi_{t_i \to t_k} - \hat{\phi}_{t_i \to t_k}\right)(\mathbf{x}) = \int_{t_i}^{t_k} \left(u_i - \hat{u}_{t_k}\right) d\tau \approx \mathbf{e} \cdot (t_k - t_i) \tag{4}$$

where $\hat{\phi}$ is the ground truth warping. Assume one refinement step is a contraction $\mathcal{U}$ as below:

$$u^{(j+1)} = \mathcal{U}\left(u^{(j)}; C_{t_k}\right) \tag{5}$$

where $C_{t_k}$ is the cost volume used in estimating optical flows. After $J$ iterations, local convergence is:

$$\left\|u^{(J)} - u_{t_k}\right\| \leq \rho^J \left\|u^{(0)} - u_{t_k}\right\| \tag{6}$$

Note that $J$ is conceptual and does not correspond to any iterative refinement module or tunable hyperparameter in BRENet, and BRENet does not perform iterative flow updates. Under the temporal smoothness of the true flow and bounded previous error $\left\|u_{t_{k-1}} - \hat{u}_{t_{k-1}}\right\| \leq \epsilon_{t_{k-1}}$, the final flow error can be represented as:

$$\|\mathbf{e}\| = \left\|u^{(J)} - u_{t_k}\right\| \leq \rho^J \left(L \cdot (t_k - t_{k-1}) + \epsilon_{t_{k-1}}\right) \tag{7}$$

where $L$ is the temporal smoothness constant. Therefore, the spatiotemporal misalignment is no longer tied to motion lag but is presented as a flow-estimation error $\mathbf{e}$, which can be optimized by $\rho^J$.

Our additional theoretical analysis presenting the Centered Kernel Alignment (CKA) (Kornblith et al., 2019) metric in Table 1 quantitatively demonstrates this advantage. Flow-RGB exhibits consistently higher CKA scores than voxel grid-RGB and 6-Channel Image-RGB across datasets, demonstrating its superior registration-centric and modality-generic capability over existing event representations.

Table 1: **Comparison of feature similarity on DDD17 and DSEC.** Higher values indicate stronger cross-modal alignment.

|                        | DDD17 | DSEC  |
| ---------------------- | ----- | ----- |
| Voxel Grid - RGB       | 0.037 | 0.009 |
| 6-Channel Image - RGB  | 0.053 | 0.016 |
| Optical flow - RGB     | 0.127 | 0.071 |

We adopt a registration-centric formulation by using optical flow as a bridge that pairs the event stream with the RGB frame's timestamp and pixel grid, providing pixel-wise correspondences. We then combine optical flow with event temporal dynamics, maintaining multi-granular receptive fields while preserving critical low-level details. Our designed bidirectional scheme further alleviates *Spatiotemporal Misalignment* by fusing forward and backward features after registration.

## 3.2 ARCHITECTURE OVERVIEW

Our proposed framework, BRENet, as illustrated in Figure 2, has three core components: a Coarse-to-Fine Estimator (CFE) for generating Motion-enhanced Event Tensors, a Bidirectional Registration Module (BRM) for multimodal fusion, and a Temporal Fusion Module (TFM) that integrates bidirectional features adaptively.

Our proposed architecture begins by processing the raw event input $E$ through a sampling stage to obtain the event frames $I_E \in \mathbb{R}^{N \times H \times W \times B}$, where $N$ is the number of frames and $B$ is the bin size. These event frames $I_E$ then undergo the flow-guided event tensorization pipeline, which comprises a flow encoder, a Temporal Convolution Module, and the proposed Coarse-to-Fine Estimator (CFE) to transform the raw event stream $E$ into a Motion-enhanced Event Tensor (MET) $M$. Note that the generated MET $\{M^f, M^b\}$ is bidirectional, consisting of forward $M^f \in \mathbb{R}^{H \times W \times C}$ and backward MET $M^b \in \mathbb{R}^{H \times W \times C}$. Simultaneously, multi-scale RGB features $F_I \in \mathbb{R}^{H \times W \times C}$ are extracted from the input image $I \in \mathbb{R}^{H \times W \times 3}$ using an image encoder. The bidirectional MET $\{M^f, M^b\}$ and the RGB features $F_I$ are then registered jointly through the Bidirectional Registration Module (BRM). The resulting forward $F_r^f \in \mathbb{R}^{H \times W \times C}$ and backward registered features $F_r^b \in \mathbb{R}^{H \times W \times C}$ are fused using the Temporal Fusion Module (TFM), which outputs the final refined feature maps $F_I' \in \mathbb{R}^{H \times W \times C}$ for the image decoder to produce the semantic segmentation masks $\hat{Y} \in \mathbb{R}^{H \times W \times 1}$. The details of each module are explained in the following subsections.

## 3.3 MOTION-ENHANCED EVENT TENSOR (MET)

**Flow-Guided Event Tensorization.** We integrate optical flow with event features, enabling the capture of continuous motion trajectories. Specifically, coarse-grained optical flows derived from a flow encoder (Gehrig et al., 2021b) capture pixel correspondences under spatiotemporal displacements, while fine-grained event features model complex temporal dependencies over all time horizons via a Temporal Convolution Module.

Given an input event stream $E$, we split it into $N$ temporal windows and sample $N$ event frames $I_E$ from these snippets. After preprocessing the event data, we first adopt a pre-trained flow encoder (Gehrig et al., 2021b) to estimate dense optical flows $\{O^f, O^b\}$ from previously sampled $N$ event frames. Optical flows $\{O^f, O^b\}$ serve as coarse-grained motion dynamics that capture global motion information. Specifically, we estimate $O^f$ based on $I_E$, and then reverse the order of all history event frames to generate backward optical flow $O^b$. Note that backward flows are computed using only observed history, and it does not involve future information beyond the current timestamp; therefore, it does not cause data leakage, nor does it prevent online deployment. Meanwhile, the Temporal Convolution Module is designed to capture event temporal features $h$, which serve as the fine-grained event features that capture the local boundary information in the context of the temporal dimension. The module consists of three sub-blocks, each containing a 3D convolutional layer with a kernel size of $2 \times 3 \times 3$, followed by a 2D convolutional layer with a kernel size of $3 \times 3$ and an average pooling layer.

**Coarse-to-Fine Estimator.** To provide a registration-centric representation and address modal discrepancy, we propose a Coarse-to-Fine Estimator (CFE). By leveraging local adaptive modeling through deformable convolution Dai et al. (2017), it effectively captures fine-grained structural details

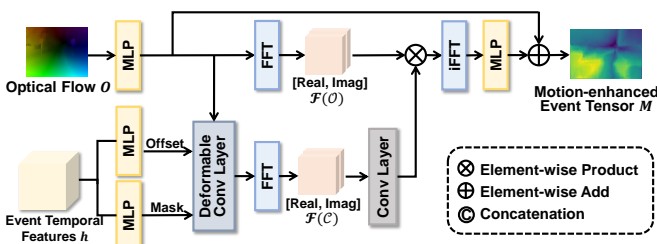

Figure 3: **Illustration of Coarse-to-Fine Estimator (CFE).**

and enhances motion-aware representations, yielding richer and more precise scene understanding for semantic segmentation.

As illustrated in Figure 3, we first apply a Multi-Layer Perceptron (MLP) (Rosenblatt, 1958) to the input optical flows $\{O^f, O^b\}$. We further employ two additional MLPs to generate the offsets and masks required by the subsequent Deformable Convolution Layer, based on event temporal features $h$. We then introduce a Deformable Convolution Layer that uses these offsets and masks, with the temporal features $h$ acting as conditions to guide adaptive sampling. This enables the kernel to adjust its spatial sampling locations based on local context, alleviating spatial misalignment through flexible receptive fields.

Next, we apply a 2D Fast Fourier Transform (FFT) (Nussbaumer & Nussbaumer, 1982) to the optical flow and convolved features, transforming them into the frequency domain. We leverage this domain based on the observation that optical flow and event representations exhibit distinct but complementary characteristics (Kim et al., 2024). We then concatenate the real and imaginary components of the FFT results to obtain a frequency representation $\mathcal{F}(\mathcal{O}) \in \mathbb{R}^{H \times \lfloor \frac{w}{2}+1 \rfloor \times 2C}$ and $\mathcal{F}(\mathcal{C}) \in \mathbb{R}^{H \times \lfloor \frac{w}{2}+1 \rfloor \times 2C}$. The final Motion-enhanced Event Tensor (MET), $M$, is obtained through element-wise multiplication, followed by MLP and a skip connection as follows:

$$M = f(\text{FFT}^{-1}(\mathcal{F}(\mathcal{O}) \otimes \mathcal{F}(\mathcal{C}))) + f(O) \tag{8}$$

where $f(\cdot)$ denotes the MLP block; $\text{FFT}^{-1}$ represents the inverse Fast Fourier Transform operation; $\mathcal{F}(\mathcal{O})$ and $\mathcal{F}(\mathcal{C})$ denotes the frequency representation (concatenation of the real and imaginary parts) of optical flow features and convolved features.

### 3.4 REGISTRATION-CENTRIC PROPAGATION

**Bidirectional Registration Module.** The registration module is flexible and adaptable to multiple network architectures. Here we follow FEVD (Kim et al., 2024) and extend their proposed Frequency-aware Cross-modal Feature Enhancement (FCFE) module into a bidirectional setting. We leverage the frequency domain in both forward and backward directions to capture low- and high-frequency components from the domain-invariant aspect. Module details can be found in the Appendix.

**Temporal Fusion Module.** The Temporal Fusion Module (TFM) fuses forward and backward registered features $F_r$ with image features $F_I$ to capture temporal coherence and enhance contextual representation across the temporal dimension. As shown in Figure 4, the module utilizes a Deformable Convolution Layer to align features from different time steps by jointly warping the same regions of bidirectional registered features $F_r$ into the input image features $F_I$:

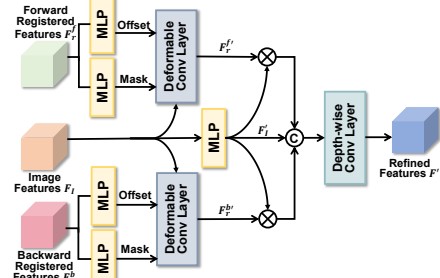

Figure 4: **Illustration of Temporal Fusion Module (TFM).**

$$F_r^{f'} = \mathcal{DC}(F_I, f_1(F_r^f), f_2(F_r^f)) \tag{9}$$

$$F_r^{b'} = \mathcal{DC}(F_I, f_3(F_r^b), f_4(F_r^b)) \tag{10}$$

where $f_n(\cdot)$ denotes different MLP blocks; $\mathcal{DC}(\cdot)$ denotes Deformable Convolution Layer. The output features are multiplied by the updated image features $F_I'$, concatenated with $F_I'$, and subsequently passed through a Depth-wise Convolution Layer (Chollet, 2017) with skip connections as follows:

$$F_I' = Concat(F_r^{f'} \otimes f(F_I), F_r^{b'} \otimes f(F_I), f(F_I)) \tag{11}$$

Table 2: **Baseline comparisons on DDD17 and DSEC dataset.** Improvements over the second-best are highlighted in green.

| Method | Modality | Backbone | Event Representation | DDD17 | | DSEC | |
|---|---|---|---|---|---|---|---|
| | | | | mIoU ↑ | mAcc ↑ | mIoU ↑ | mAcc ↑ |
| SegFormer [NeurIPS21] | RGB | MiT-B2 | — | 71.05 | 95.73 | 71.99 | 94.97 |
| SegNeXt [NeurIPS22] | RGB | MSCAN-B | — | 71.46 | 95.97 | 71.55 | 94.89 |
| EV-SegNet [CVPR19] | Event | Xception | 6-Channel Image | 54.81 | 89.76 | 51.76 | 88.61 |
| ESS [ECCV22] | Event | E2ViD | Voxel Grid | 61.37 | 91.08 | 51.57 | 89.25 |
| EDCNet-S2D [TITS22] | RGB-Event | ResNet-101 | Voxel Grid | 61.99 | 93.80 | 56.75 | 92.39 |
| CMX [TITS23] | RGB-Event | MiT-B2 | Voxel Grid | 67.47 | 94.20 | 65.29 | 92.61 |
| CMNeXt [CVPR23] | RGB-Event | MiT-B2 | Voxel Grid | 66.99 | 93.82 | 67.2 | 93.13 |
| HALSIE [WACV24] | RGB-Event | FCN | Voxel Grid | 60.66 | 92.50 | 52.43 | 89.01 |
| CMESS [RAL24] | RGB-Event | E2ViD | Voxel Grid | 64.30 | 92.07 | 59.53 | 91.11 |
| OpenESS [CVPR24] | RGB-Event | E2VID | Voxel Grid | 63.00 | 91.05 | 57.21 | 90.21 |
| SpikingEDN [TNNLS24] | RGB-Event | FCN | Voxel Grid | 72.57 | - | 58.32 | - |
| SE-Adapter [ICRA24] | RGB-Event | SAM | MSP | 69.06 | 95.32 | 69.77 | 93.58 |
| EISNet [TMM24] | RGB-Event | MiT-B2 | AET | 75.03 | 96.04 | 73.07 | 95.12 |
| Spike-BRGNet [TCSVT25] | RGB-Event | MiT-B2 | Voxel Grid | 54.72 | - | 54.95 | - |
| Hybrid-Seg [AAAI25] | RGB-Event | FCN | Voxel Grid | 67.31 | 95.07 | 66.57 | 94.27 |
| **BRENet (Ours)** | RGB-Event | MiT-B2 | MET | **78.56** (+3.53) | **96.61** (+0.57) | **74.94** (+1.87) | **95.85** (+0.73) |

$$\boldsymbol{F}' = \mathcal{DWC}(\boldsymbol{F}'_{\boldsymbol{I}}) + \boldsymbol{F}_{\boldsymbol{I}} \tag{12}$$

where $f(\cdot)$ denotes MLP block and $\mathcal{DWC}(\cdot)$ denotes the Depth-wise Convolution Layer. The output of TFM is refined features $\boldsymbol{F}'_{\boldsymbol{I}}$.

Finally, BRENet adopts a lightweight image decoder, consisting of one MLP block, to predict segmentation results $\hat{\boldsymbol{Y}}$.

## 4 EXPERIMENTS

### 4.1 IMPLEMENTATION DETAILS

**Training details.** We implement our work using PyTorch (Paszke, 2019) and MMSeg (Contributors, 2020). The loss function is the per-pixel cross-entropy loss, following common practice with the Online Hard Example Mining strategy. We train the model using the AdamW optimizer (Loshchilov & Hutter, 2017) and a polynomial learning rate schedule with an initial learning rate of 6e-5. For the flow encoder, we adopt E-RAFT Gehrig et al. (2021b), which is an efficient, event-based optical flow estimation framework pre-trained on the DSEC dataset Gehrig et al. (2021a). E-RAFT is trained strictly on the training split of DSEC, which is disjoint from the testing set used in our segmentation experiments This practice is consistent with standard methodologies in event-based optical flow and does not constitute data leakage. We fine-tune the flow encoder while training on other datasets. We use 2 NVIDIA RTX A5000 GPUs for training. All of the models are trained for 80k iterations. More details are in the Appendix.

### 4.2 DATASETS

We used 4 public large-scale datasets : **DDD17** (Binas et al., 2017), **DSEC** (Gehrig et al., 2021a), **DELIVER** Zhang et al. (2023b) and **M3ED** (Chaney et al., 2023). DDD17 contains 15950 grey-scale images for training and 3890 images for testing with 6 categories. DSEC contains 11 video sequences (10891 frames) with 11 categories. DELIVER is for RGB-X semantic segmentation, and we evaluated our model on it for robustness. For M3ED, we chose 4 sequences (5516 images) for training and 2 sequences (2481 images) for testing from the Urban Day subset with manual inspections.

### 4.3 QUANTITATIVE RESULTS

We evaluate BRENet and SOTA models on the DDD17 (Binas et al., 2017) and DSEC (Gehrig et al., 2021a) datasets in Table 2. SOTA models evaluated include RGB-based models (SegFormer (Xie et al., 2021), SegNeXt (Guo et al., 2022)), Event-based models (EV-SegNet(Alonso & Murillo, 2019), ESS (Sun et al., 2022)), and RGB-Event models (EDCNet-S2D (Zhang et al., 2021), CMX (Zhang et al., 2023a), CMNeXt (Zhang et al., 2023b), HALSIE (Das Biswas et al., 2024), CMESS (Xie et al., 2024b), OpenESS (Kong et al., 2024), SpikingEDN (Zhang et al., 2024), SE-Adapter (Yao et al.,

Table 3: **Baseline comparisons on DELIVER and M3ED datasets using mIoU.**

| Method | DELIVER | M3ED |
|---|---|---|
| CMX [TITS23] | 56.37 | 52.64 |
| CMNeXt [CVPE23] | 58.04 | 53.85 |
| Any2Seg [ECCV24] | 57.83 | - |
| GeminiFusion [ICML24] | 58.50 | - |
| EISNet [TMM24] | - | 59.91 |
| **BRENet (Ours)** | **63.13** (+4.63) | **67.28** (+7.37) |

Table 4: **Model complexity on DDD17 dataset.**

| Method | Params (M)↓ | GFLOPs ↓ | Latency (ms)↓ | mIoU↑ |
|---|---|---|---|---|
| EV-SegNet | 29.09 | - | 143.7 | 51.76 |
| CMX | 66.56 | 64.9 | 104.2 | 67.47 |
| CMNeXt | 58.68 | 65.3 | 101.7 | 66.99 |
| EISNet | **34.39** | 69.2 | **83.5** | 75.03 |
| BRENet (Ours) | 37.69 | **55.2** | 94.3 | **78.56** |

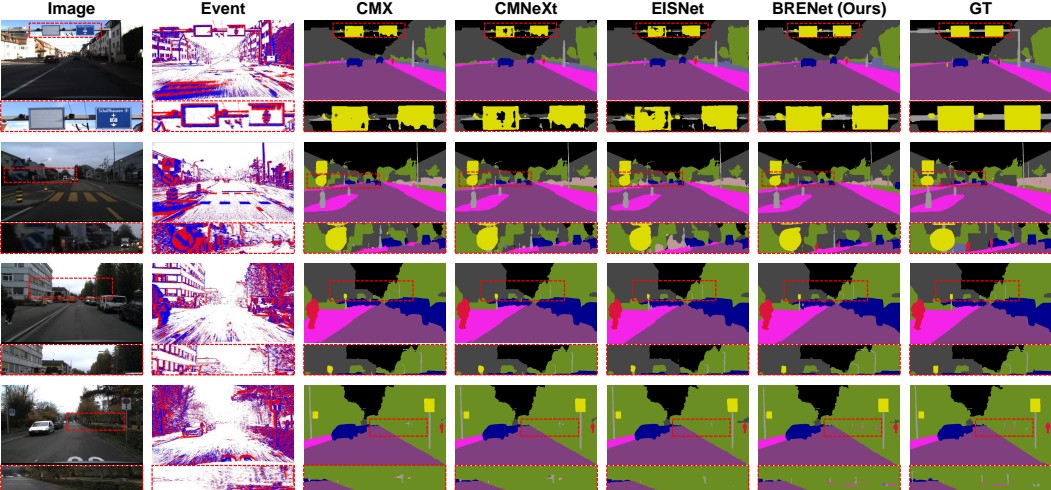

Figure 5: **Qualitative results on DSEC dataset.** The proposed BRENet produces images with enhanced boundary details and more robust predictions compared to SOTA methods. More qualitative results are in the Appendix.

2024), EISNet (Xie et al., 2024a), Spike-BRGNet Long et al. (2024), Hybrid-Seg Li et al. (2025), Any2Seg Zheng et al. (2024), GeminiFusion Jia et al. (2024)) using their default settings.

As shown in Table 2, BRENet achieves 78.56% mIoU and 96.61% mAcc on DDD17 and 74.94% mIoU and 95.85% mAcc on DSEC using the MiT-B2 backbone (Xie et al., 2021). It outperforms SOTA methods on both datasets by a large margin. We present results exclusively with MiT-B2 for direct comparison, as most SOTA models adopt this backbone. We further evaluate model performance using the DELIVER Zhang et al. (2023b) and M3ED (Chaney et al., 2023) datasets in Table 3. The table indicates that BRENet achieves 63.13% and 67.28% mIoU on DELIVER and M3ED, respectively. Our model achieves the best performance on all datasets and significantly outperforms SOTA methods, improving 7.37% mIoU on M3ED, 4.63% mIoU on DELIVER, 3.53% mIoU on DDD17, and 1.87% mIoU on DSEC compared to the previous best models. It is noteworthy that our BRENet demonstrates substantial performance improvements over the SAM-based model, SE-Adapter (Yao et al., 2024). Although the flow encoder, E-RAFT, was originally trained on DSEC, the largest gains of BRENet occur on three other datasets (+3.53 mIoU on DDD17, +4.63 mIoU on DELIVER, and +7.37 mIoU on M3ED) besides DSEC, where no such prior advantage exists.

Additionally, we analyze model complexity on the DDD17 dataset in Table 4 on one single A5000 GPU. While earlier approaches exhibit the smallest parameter size, they achieve substantially lower mIoU. In contrast, our method achieves SOTA performance (78.56% mIoU) while having a similar model size and MACs comparable to recent high-performing approaches. Although our bidirectional propagation slightly increases overhead, our efficient flow encoder ensures comparable processing latency. The results demonstrate that our method effectively balances the trade-off between accuracy and computational efficiency, offering a more practical solution for real-world event-based vision applications.

## 4.4 QUALITATIVE RESULTS

We compare the qualitative results of BRENet with SOTA models (e.g., CMX (Zhang et al., 2023a), CMNeXt (Zhang et al., 2023b), and EISNet (Xie et al., 2024a)) using their default settings in Figure

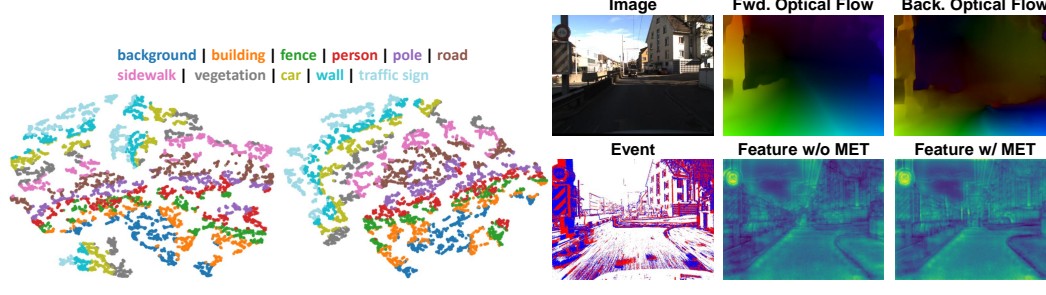

(a) RGB      (b) RGB w/ MET

Figure 6: **Effects of adding MET visualized with t-SNE.**

Figure 7: **Visualization of forward and backward optical flows and feature maps adding MET.** Best viewed with zoom.

Table 5: **Ablation study on DDD17 dataset.**

| Variant | Event Representation | Bi-Propa. | RM | TFM | mIoU ↑ |
|---------|----------------------|-----------|-----|------|--------|
| 1 | — | | | | 71.05 |
| 2 | Voxel Grid | | | | 72.82 |
| 3 | AET | | | | 73.64 |
| 4 | Optical Flow | | | | 72.96 |
| 5 | Event Temporal Features | | | | 73.27 |
| 6 | MET | | | | **74.32** |
| 7 | MET | | ✓ | ✓ | 76.94 |
| 8 | MET | ✓ | | ✓ | 77.61 |
| 9 | MET | ✓ | ✓ | | 77.13 |
| 10 | MET | ✓ | ✓ | ✓ | **78.56** |

Table 6: **Ablation study on plug-and-play performance of MET in SOTA methods using mIoU.**

| Method | DDD17 | DSEC |
|--------|-------|------|
| CMX | 67.47 | 65.29 |
| CMX + MET | 71.18 (+3.71) | 68.34 (+3.05) |
| EISNet | 75.03 | 73.07 |
| EISNet + MET | 75.64 (+0.61) | 73.56 (+0.49) |

5. We analyze qualitative results across diverse scenarios, including varying lighting conditions (rows 1-2) and crowded scenes (rows 3-4). These models struggle to capture small moving objects (e.g., people in row 2) and locate accurate boundaries between different categories (e.g., road signs in row 1), while BRENet enhances boundary accuracy and effectively resolves the blurring in fast motion.

We additionally present the t-SNE visualization of image features after adding MET in Figure 6. In contrast to (a), we observe more distinct and well-separated clusters in (b), indicating enhanced feature differentiation. This suggests that incorporating MET improves the model's ability to learn more discriminative features, further enhancing its performance.

To further analyze the practical benefits of MET, we visualize features in Figure 7. It shows that incorporating MET selectively highlights areas with rich semantic information (e.g., buildings) while reducing attention to less important regions (e.g., sidewalks). MET's multi-granular receptive fields preserve low-level details with motion semantics, leading to clearer boundary details with less noise.

## 4.5 ABLATION STUDY

**Design choices for event representations.** To validate the impact of different event representations on models, we compare our proposed MET with commonly used event representations, e.g., voxel grid (Zhu et al., 2019) and AET (Xie et al., 2024a). We take different event representations as input to the baseline and concatenate the event features with the image features in the intermediate layer. The corresponding results are listed in rows 1-6 of Table 5. Our proposed MET outperforms other SOTA representations by 2.06% (voxel grid) and 0.92% (AET). Row 4-6 provides a granularity analysis of our design, showing that both coarse-grained and fine-grained features are important. With a single input (variants 4&5), event temporal features improve more, enhancing low-level details. Adding optical flow (variant 6) restores high-level motion and semantics.

**Design choices for proposed modules.** To validate the effectiveness of each component, we further evaluate four variants (7-10) of BRENet. Specifically, they are: (7) removing bidirectional propagation, (8) removing BRM, (9) removing TFM and applying concatenation, and (10) employing all designed components. Results are summarized in Table 5. Without bidirectional propagation, performance drastically drops (comparing variants 7 & 10). This is because adding bidirectional contexts and extending RM to bidirectional RM enhances motion dynamics in challenging scenarios.

Table 7: **Ablation study of event bin number $N$ on DSEC dataset.**

| Method | Event Bin $N$ | mIoU ↑ | mAcc ↑ |
|--------|---------------|--------|--------|
| BRENet | 1 | 74.2 | 95.68 |
| BRENet | 3 | 74.66 | 95.81 |
| BRENet | 5 | 74.67 | 95.7 |
| BRENet | 15 | **74.94** | **95.85** |

Table 8: **Ablation study of different flow encoders on DSEC dataset.**

| Method | mIoU ↑ |
|--------|--------|
| BRENet + E-RAFT [3DV21] | 74.94 |
| BRENet + TMA [ICCV23] | 74.62 |
| BRENet + ADMFlow [ICCV23] | 75.11 |
| BRENet + EEMFlow [CVPR24] | 75.36 |

Incorporating BRM (variant 8) and TFM (variant 9) improves results, indicating that they can successfully mitigate temporal and spatial misalignments.

**Plug-and-play performance of MET.** We provide additional results of incorporating MET into SOTA methods, CMX (Zhang et al., 2023a) and EISNet (Xie et al., 2024a) in Table 6. We observe that CMX with MET achieves 71.18% and 68.34% mIoU on DDD17 and DSEC, which improves the original method by 3.71% and 3.05%. Similarly, EISNet with MET achieves 75.64% and 73.56% mIoU on DDD17 and DSEC, improving the original architecture by 0.61% and 0.49%.

**Selection of event bin size.** We also perform ablation studies on the DSEC dataset Gehrig et al. (2021a) to investigate the impact of different event bin sizes. Results in Table 7 indicate that increasing $N$ likely contributes to better learning of temporal dynamics. However, the improvement is modest and not decisive. Therefore, the performance gain isn't mainly from the larger number of event bins but from our unique design of MET and subsequent modules.

**Selection of different flow encoders.** We intentionally adopt E-RAFT Gehrig et al. (2021b) as our default choice to avoid utilizing the latest advancements to demonstrate the robustness and generalizability of our approach without relying on cutting-edge flow quality. To thoroughly analyze the robustness, we evaluate BRENet with various flow encoders. As shown in Table 8, our model achieves stable performance using ADM-Flow Luo et al. (2023) and EEMFlow Luo et al. (2024). While adding TMA Liu et al. (2023) achieves a lower mIoU, it still significantly outperforms SOTA models, e.g., EISNet Xie et al. (2024a). It demonstrates that MET maintains stable performance across various flow backbones, highlighting that our model does not rely on high-quality optical flow but instead benefits from the inherent structural registration. It shows that optical flows are not utilized for pixel-level precision of feature aggregation but rather as a bridge and coarse visual guidance for registration. Specifically, it provides flow-aware cues to temporally align asynchronous event streams and convert sparse events into dense, motion-enhanced representations. It shows that our approach is not sensitive to potential flow errors and remains effective under different flow qualities, even when it is imprecise.

## 5 CONCLUSION

In this paper, we recast RGB-Event semantic segmentation as a registration problem rather than fusion. Our proposed framework, BRENet, establishes pixel-wise correspondences through flow-guided bidirectional registration and then fuses aligned features. We further introduce a Motion-enhanced Event Tensor (MET) representation, which converts asynchronous, sparse events into a dense, temporally coherent representation by combining coarse optical flows with fine event temporal features. Bidirectional registration and MET effectively capture temporal context information while bridging modality gaps, mitigating *Spatiotemporal* and *Modal Misalignment*. Experimental results on DDD17, DSEC, DELIVER, and M3ED demonstrate the effectiveness of our model. Our findings and *registers first, then fuses* design ideology suggest a promising direction for future research in RGB-Event perception.

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

## APPENDIX OVERVIEW

In this Appendix, we provide additional details to compliment the content of the paper, including the model details of the BRM (Section A), additional visualization (Section B), additional ablation studies (Section C), analysis of the results from comparative methods (Section D), Misalignment visualization (Section F), and limitation of proposed method (Section G).

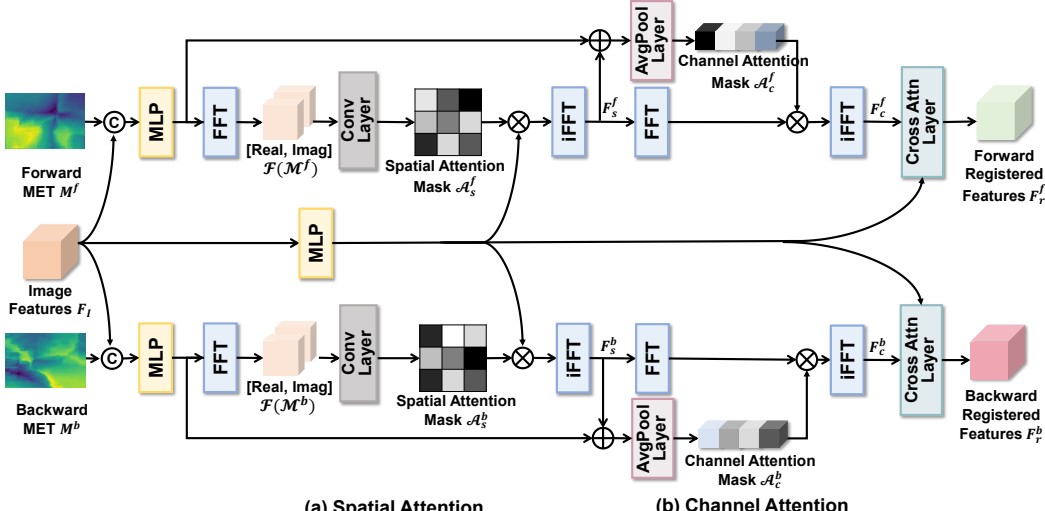

Figure 8: **Illustration of Bidirectional Registration Module (BRM).** From left to right, the components are Spatial and Channel Attention Blocks.

## A MORE DETAILS

### A.1 ARCHITECTURAL DETAILS

#### A.1.1 BIDIRECTIONAL REGISTRATION MODULE

Inspired by FEVD Kim et al. (2024), we build the Bidirectional Registration Module (BRM) where we leverage the frequency domain in both the forward and backward directions to mitigate the misalignment between RGB features and MET features. It consists of two main components: the Spatial Attention Block and Channel Attention Block.

As shown in Figure 8, we first concatenate the image feature $\boldsymbol{F_I}$ from the image encoder and the bidirectional METs $\{\boldsymbol{M^f}, \boldsymbol{M^b}\}$, and apply an MLP to each modality feature.

For the Spatial Attention Block, we apply 2D Fast Fourier Transform (FFT) to MET features. The generated real and imaginary components are concatenated to preserve both amplitude and phase information as follows:

$$\mathcal{F}(\mathcal{M}) = \text{FFT}(f(Concat(\boldsymbol{M}, \boldsymbol{F_I}))) \tag{13}$$

where $\mathcal{F}(\mathcal{M})$ can be viewed as the frequency representation of MET Features in both directions and FFT$(\cdot)$ includes Fast Fourier Transform operation, followed by concatenation. We then adopt a Convolution Layer for generating the Spatial Attention Mask, and the proposed Spatial Attention Block is calculated as:

$$\mathcal{A}_s = \sigma(\mathcal{C}_{3\times3}(\mathcal{F}(\mathcal{M}))) \tag{14}$$

$$\boldsymbol{F_s} = \text{FFT}^{-1}(\mathcal{A}_s \otimes f(\boldsymbol{F_I})) \tag{15}$$

where $\mathcal{C}_{3\times3}(\cdot)$ represents Convolution Layer with kernel size $3 \times 3$, followed by ReLU and Sigmoid functions; $\sigma(\cdot)$ indicates Sigmoid activation function; FFT$^{-1}(\cdot)$ refers to inverse Fast Fourier Transform operation. Note that the Spatial Attention Block is applied for both forward and backward directions, generating spatial correlated features $\{\boldsymbol{F_s^f}, \boldsymbol{F_s^b}\} \in \mathbb{R}^{H \times W \times C}$.

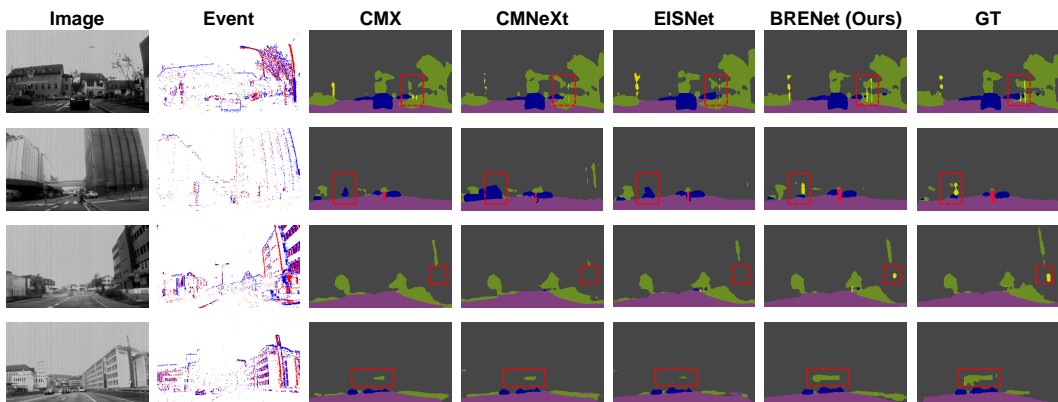

Figure 9: **Qualitative results on DDD17 dataset.** It shows that our model generates more robust and temporally consistent results than SOTA methods.

For Channel Attention Block, we similarly apply 2D Fast Fourier Transform (FFT) Nussbaumer & Nussbaumer (1982) for spatially correlated features $\boldsymbol{F_s}$. The Channel Attention Mask and the following channel correlated features $\boldsymbol{F_c}$ are calculated based on the Average Pooling Layer as:

$$\mathcal{A}_c = \mathcal{AP}(\boldsymbol{F_s} \oplus f(Concat(\boldsymbol{M}, \boldsymbol{F_I}))) \tag{16}$$

$$\boldsymbol{F_c} = \text{FFT}^{-1}(\mathcal{A}_c \otimes \text{FFT}(\boldsymbol{F_s})) \tag{17}$$

where $\mathcal{AP}(\cdot)$ represents the Average Pooling Layer with 2D Adaptive Average Pooling, followed by ReLU and Sigmoid functions.

An additional Cross-attention Layer is employed to generate the final registered features $\boldsymbol{F_r^f}$ and $\boldsymbol{F_r^b}$ in both directions, where image features act as queries and channel correlated features $\boldsymbol{F_c}$ serve as keys/values.

### A.1.2 DECODER

We employ a lightweight MLP decoder that significantly reduces high computational costs compared with existing methods. The key to enabling such a simple decoder is that our proposed MET representations and bidirectional mechanism alleviate spatiotemporal and modal misalignments, leading to better feature extraction ability.

### A.2 OTHER IMPLEMENTATION DETAILS

**Training details:** We use MiT-B2 from SegFormer Xie et al. (2021) pre-trained on the ImageNet-1K dataset Deng et al. (2009) as the backbone, following the common practice of most SOTA methods. The flow encoder, E-RAFT, is a lightweight design with only 5M parameters and is trained on 7,000 images.

**Data preprocessing.** The data augmentation used in our work includes random cropping, random flipping, and photometric distortion, following Xie et al. (2024a). During training, we crop the RGB images to $512 \times 512$ solely from the M3ED dataset Chaney et al. (2023). We do not explore random cropping on the other 2 datasets since their input resolution is much smaller.

**Evaluation metrics.** We use mean Intersection over Union (mIoU) and pixel accuracy (mAcc) to measure segmentation performance. The model complexity is measured by parameter size and MACs (Patil & Kulkarni, 2018).

## B ADDITIONAL VISUALIZATION

### B.1 ADDITIONAL QUALITATIVE RESULTS

To further validate our model, we present a visualization of segmentation predictions on several samples from the DDD17 Binas et al. (2017) dataset to highlight the robustness of our approach,

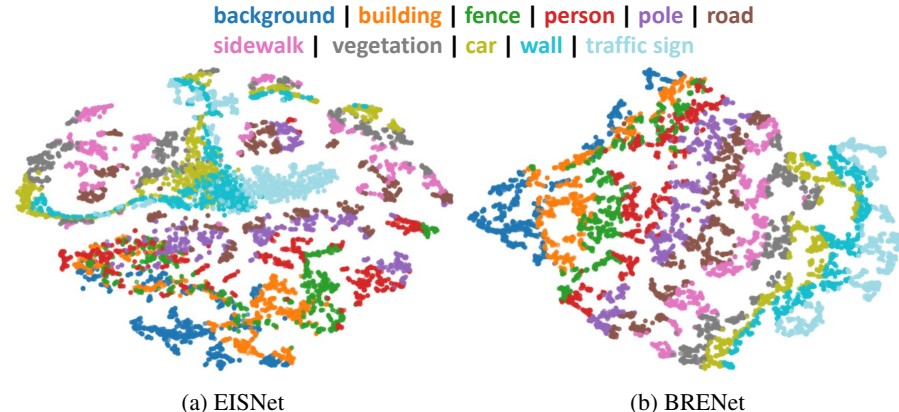

(a) EISNet                                    (b) BRENet

Figure 10: **Comparisons between EISNet and BRENet via t-SNE visualisation.** Best viewed with zoom.

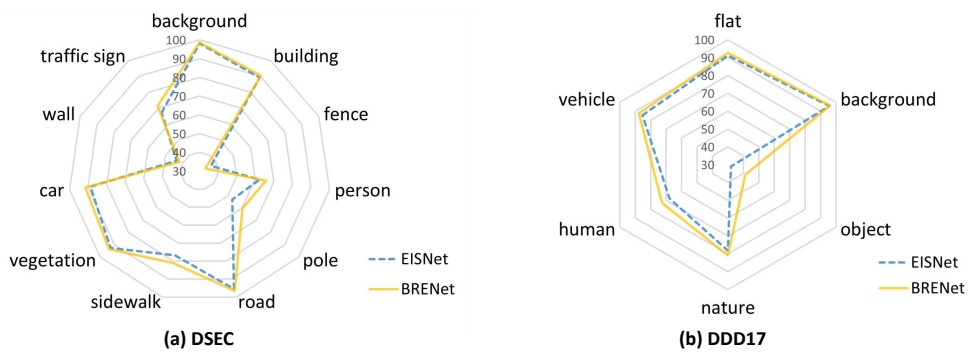

(a) DSEC                                    (b) DDD17

Figure 11: **Per-class comparisons with SOTA method on DSEC and DDD17 datasets.**

as shown in Figure 9. Specifically, we compare the qualitative results of our method with the baseline EISNet Xie et al. (2024a), utilizing its trained weights. In the predictions generated by EISNet, some small objects are omitted, such as traffic signs in rows 1-3 and trees in row 4. This indicates its limitations in accurately recognizing dynamic motions and in reducing blurring effects. In contrast, BRENet captures smaller objects, preserves intricate details, and generates sharper and more precise object boundaries. These samples showcase BRENet's ability to tackle challenging scenarios involving rapid motion and complex visual environments, effectively addressing limitations in prior methods.

### B.2 ADDITIONAL T-SNE VISUALIZATION

Figure 10 depicts the t-SNE visualization of the feature maps taken before the prediction head for both EISNet and the proposed BRENet on the DSEC Gehrig et al. (2021a) dataset. Whereas EISNet features form partially overlapping groups, BRENet produces more distinct and well-grouped clusters, indicating stronger class discriminability and robustness—properties that align with its superior quantitative performance.

### B.3 PER-CLASS COMPARISON

We present a comprehensive analysis across all classes in Figure 11 on DSEC Gehrig et al. (2021a) and DDD17 Binas et al. (2017), focusing on per-class mIoU performance. BRENet outperforms EISNet in every category on DDD17 Binas et al. (2017) and in all but the "fence" & "wall" classes on DSEC Gehrig et al. (2021a), where the scores are comparable. It demonstrates BRENet's balanced and robust performance across classes, achieving high mIoU scores in previously challenging areas.

Table 9: **Ablation study of our model with different backbones on DSEC dataset.**

| Backbone | Params(M) ↓ | mIoU↑ | mAcc ↑ |
|---|---|---|---|
| MiT-B0 | 16.68 | 68.76 | 94.79 |
| MiT-B2 | 37.69 | 74.94 | 95.85 |
| MiT-B5 | 94.93 | 75.22 | 95.88 |

Table 10: **Ablation study on DSEC dataset.**

| Variant | Event Repre. | Bi-Propa. | RM | TFM | mIoU ↑ |
|---|---|---|---|---|---|
| 1 | — | | | | 71.99 |
| 2 | Voxel Grid | | | | 72.36 |
| 3 | AET | | | | 72.84 |
| 4 | Optical Flow | | | | 72.52 |
| 5 | Event Temporal Features | | | | 72.95 |
| 6 | MET | | | | **73.24** |
| 7 | MET | | ✓ | ✓ | 73.97 |
| 8 | MET | ✓ | | ✓ | 74.27 |
| 9 | MET | ✓ | ✓ | | 74.39 |
| 10 | MET | ✓ | ✓ | ✓ | **74.94** |

We attribute the leading performance of BRENet to the following three factors: (1) Our proposed MET transforms sparse events into visual-based tensors with optical flows, alleviating modal misalignment. (2) The bidirectional feature propagation leverages temporal coherence in both forward and backward directions to improve spatiotemporal consistency. (3) The TFM adaptively aligns bidirectional features, further addressing the spatial misalignment issue.

## C    ADDITIONAL ABLATION STUDY

**Selection of different backbones.** In addition, we validate our model using different backbones, as summarized in Table 9. Specifically, we use MiT-B0, MiT-B2, and MiT-B5 to analyze the impact of varying network capacities on performance. Leveraging more powerful backbones consistently improves the results. Replacing MiT-B0 with MiT-B2 leads to a significant mIoU increase of 9.0%. Similarly, replacing MiT-B2 with MiT-B5 yields a further mIoU improvement of 0.4%, accompanied by a 151.9% increase in model size. After employing the most powerful backbone MiT-B5, BRENet achieves 75.22% mIoU, outperforming SOTA methods by 2.9%.

**Design choices for the DSEC dataset.** Furthermore, we validate the designs of MET and subsequent modules on the DSEC Gehrig et al. (2021a) dataset in Table 10. Variant 6, which employs MET, achieves 73.24% mIoU and outperforms other event representations. To validate the effectiveness of each component, we further evaluate four variants (7-10) of BRENet. Specifically, they are: (i) removing bidirectional propagation, (ii) removing BRM, (iii) removing TFM and applying concatenation, and (iv) employing all designed components. Variant 10, equipped with all proposed modules, improves SOTA performance by 2.6% mIoU. Bidirectional propagation, RM, and TFM contribute 1.3%, 0.9%, and 0.7% mIoU improvements, respectively. From these findings, we draw a similar conclusion: combining MET with bidirectional propagation, RM, and TFM effectively mitigates misalignments and enhances spatiotemporal coherence.

## D    PERFORMANCE VS. MODEL SIZE

We further conduct the Performance vs. Model Size analysis on the DDD17 dataset Binas et al. (2017) in Figure 12. The results demonstrate that BRENet achieves significant improvements over state-of-

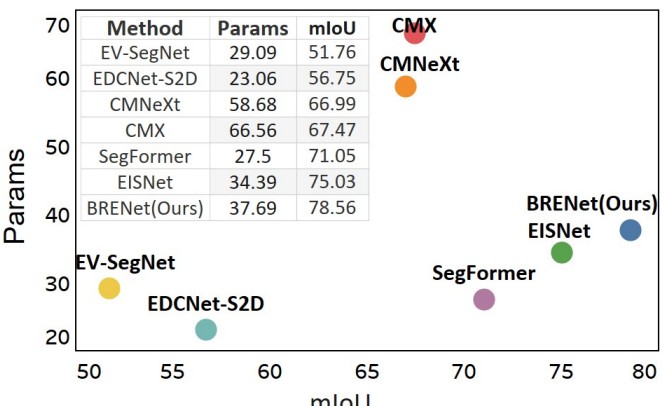

Figure 12: **Performance vs. model size on DDD17 dataset.**

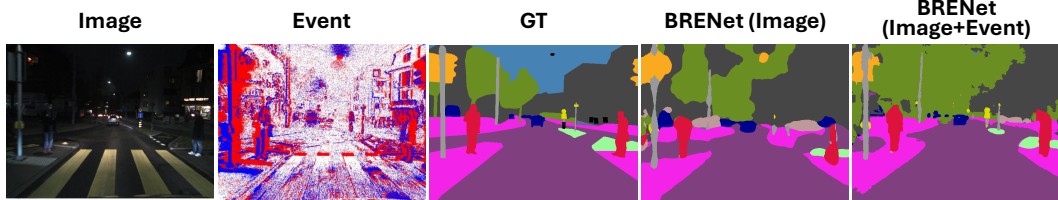

Figure 13: **Visualization on DSEC-Night testing set.**

the-art (SOTA) methods in segmentation performance while maintaining a comparable or slightly increased model size. Specifically, BRENet increases the parameter size by approximately 13M compared to the smallest method, EDCNet-S2D Bazazian & Parés (2021), and only 3M more than the most recent approach, EISNet Xie et al. (2024a). Despite these modest increases in model size, BRENet achieves a remarkable mIoU improvement of 4.7%. This analysis demonstrates BRENet's efficiency in balancing performance gains with computational costs.

## E  FAILURE CASE

We perform zero-shot evaluations on DSEC-Night Gehrig et al. (2021a) for challenging low-light scenarios. These experiments reveal that our model maintains strong segmentation accuracy even when flow estimation becomes noisy, as shown in Figure 13. Specifically, in moderately dark regions, events remain robust and support reliable flow estimation to guide RGB features. In extremely dark regions, event data still preserve subtle motion cues, while optical flows naturally and significantly degrade. As a result, predictions in these areas become more ambiguous, particularly for fine-grained boundaries (e.g., tree vs. sky).

## F  MISALIGNMENT VISUALIZATION

As discussed in Section 1, RGB-Event fusion suffers from two major misalignments: **Spatiotemporal** and **Modal Misalignment**. Figure 14 illustrates both types of misalignment in the DSEC dataset. On one hand, asynchronous sampling causes temporal lag and spatial shifts, resulting in noticeable boundary mismatches, especially in fast motion scenarios, e.g., driving. Such spatial shifts make pixel-wise predictions more challenging. On the other hand, the event modality differs from RGB; it only records brightness changes, generating sparse, edge-focused scattered points resembling point clouds. In contrast, RGB frames provide dense appearance and semantic contexts. This disparity leads to a significant representation gap, making feature alignment across modalities difficult.

## G  LIMITATION

Although BRENet attains similar latency compared to SOTA models owing to its efficient flow encoder and TFM design, bidirectional propagation and deformable convolutions still add non-trivial computational costs. Consequently, the current model is not yet suitable for edge devices with limited computational resources. Developing a lightweight variant that preserves accuracy under such hardware budgets is an intriguing topic for future work.

## H  THE USE OF LARGE LANGUAGE MODELS (LLMS)

We used LLMs as a general-purpose assistive tool to improve the writing. Specifically, it helped to check for typos, grammar and style issues, and minor notation inconsistencies. It also suggested alternative phrasings for clarity. The LLM did not propose research ideas or design models. All LLM suggestions were manually reviewed before their incorporation.

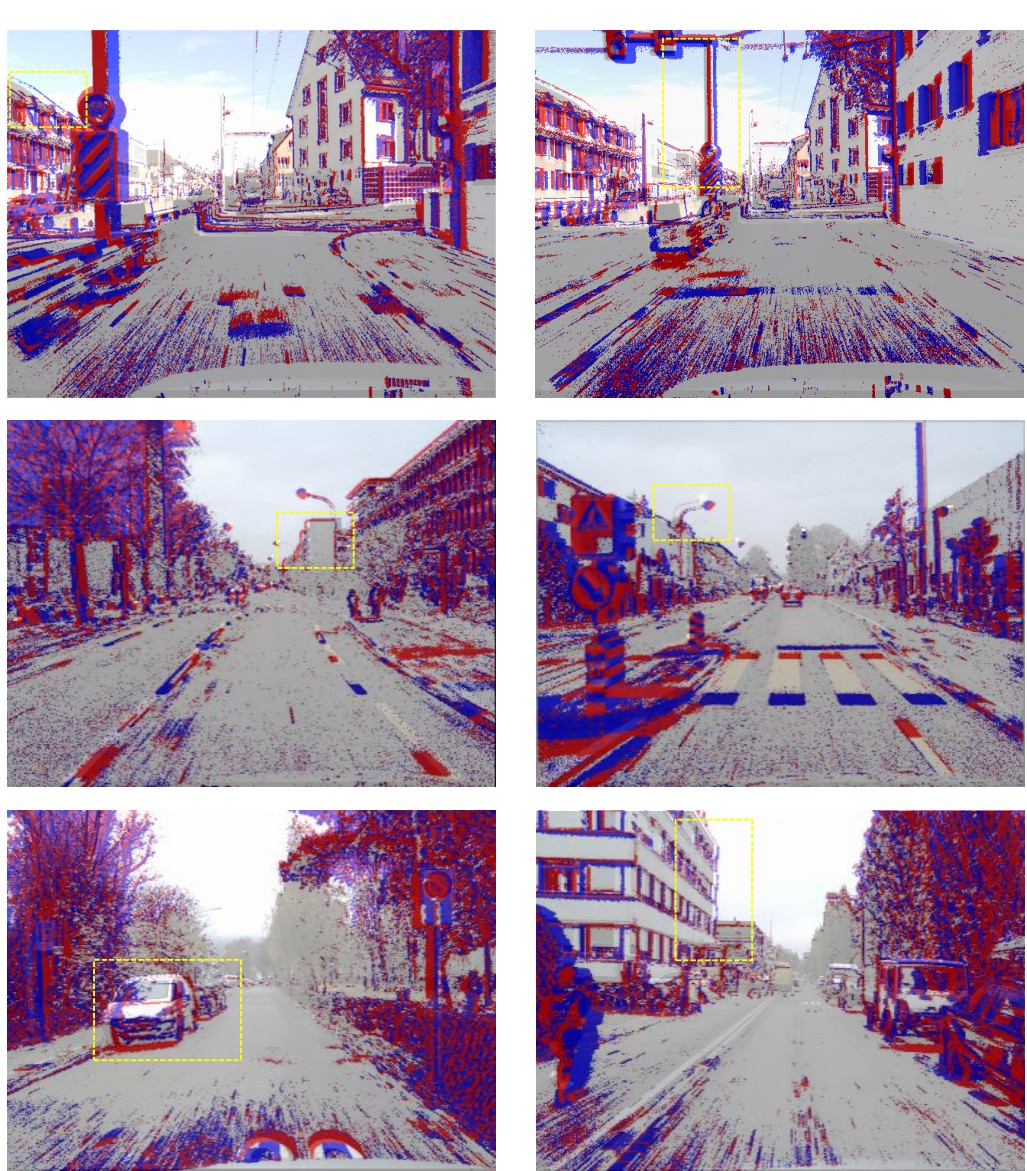

Figure 14: **Misalignment visualization.** The scattered events exhibit a clear domain gap and obvious spatial shifts due to different temporal resolution, compared to the RGB modality.

