# OpenReview forum: "Learning Flow-Guided Registration for RGB–Event Semantic Segmentation"
_ICLR.cc/2026/Conference — Submitted to ICLR 2026_

### Official Review · Reviewer_ESLK · 2025-10-26

**Soundness:** 3
**Presentation:** 3
**Contribution:** 3
**Rating:** 4
**Confidence:** 2

**Summary:**

This work proposes a method for semantic segmentation with RGB + event camera inputs. It first aligns the feature of event data and rgb images using optical flow on the event samples, then use the aligned feature to assist the final rgb image segmentation. Experiment shows improvements on finetuned datasets over previous RGB + event camera work.

**Strengths:**

The idea of using optical flow to align event data with RGB data is interesting. The authors claim that it works better than previous fusion-based methods, where the proposed one finds better pixel-level correspondences.

The proposed method effectively aligns event data with natural images, leading to better representation and performance improvement over previous RGB+event methods on several datasets.

**Weaknesses:**

1. It is unclear from both method description and figure explanation why the proposed method is not a "fusion-based" method. I understand it performs alignment step, but ultimately it is still a way to fuse 2 modalities. How does existing methods fuse the information should be presented clearer in fig 1 (a).
2. The RGB-based baselines in table 2 are out-dated, there are many more recent methods that perform (much) better, which should be compared against. This also raises a serious concern: though theoretically the method is complimentary with RGB-based strategies, is adding event data for RGB segmentation really a good choice comparing to simply scaling up the training data in the RGB side, which is arguably much easier and should provide much better zero-shot generalization capabilities. The complex design of the proposed method seems to be only marginally better than RGB-based counter-parts on some metrics (accuracy) and datasets (DSEC).

**Questions:**

Are the numbers of the RGB methods your own implementation or the results reported in the original paper?

---

> ### Author Response · Authors · 2025-11-19
> **(1/2) Official Comment by Authors**
>
> We sincerely thank the reviewer for the constructive feedback. We greatly appreciate your recognition of the paper’s strengths, including its novelty and performance improvement. Your comments are invaluable and will guide us in further improving the clarity and impact of our work. We address each of the raised concerns in detail below:
>
> > Fusion-centric vs registration-centric.
>
> We thank the reviewer for raising this important point for our claimed contribution. We clarify the conceptual and architectural difference between *registration-centric* and *fusion-centric* frameworks below.
>
> **(1) Fusion-based methods assume pre-aligned modalities and fuse features directly.**
>
> Prior RGB–Event works (e.g., CMX, CMNeXt, and EISNet) follow a **fusion-only** paradigm:
>
> - The RGB and event streams are fused in feature space via various fusion modules, e.g., attention-based and gating-based modules.
> - These methods implicitly assume **spatial and temporal alignment** between the two modalities without considering any registration steps.
> - Misalignment is treated as noise and is *not corrected* before fusion.
>
> Thus, these pipelines fuse information under the assumption that both modalities already correspond pixel-to-pixel.
>
> **(2) BRENet explicitly converts RGB-Event perception into a *registration problem* before fusion.**
>
> In contrast, BRENet performs **explicit cross-modal registration *before* fusion**:
>
> - We use optical flow as a **bridge** to compute spatial forward/backward correspondence fields.
> - We warp event features from a sequence past timestamps onto the current RGB frame to reduce temporal misalignment.
> - Only **after this explicit spatiotemporal registration**, TFM fuses the aligned RGB and event representations.
>
> Therefore, fusion is *not the starting point*, but the **final aggregation step after registration**. BRENet differs in both motivation and operational sequence:
>
> Fusion-based: fuse → no feature alignment
>
> **BRENet (ours): register → fuse only after alignment**
>
> This design fundamentally changes how the modalities interact.
>
> **(3) Why BRENet is not simply “a fusion method that happens to do alignment.”**
>
> The key conceptual shift is that *alignment is not treated as an auxiliary enhancement*, but as the **core objective** of the architecture:
>
> - MET transforms events into a temporally dense, warp-ready space.
> - BRM aligns RGB and events via bidirectional registration.
> - TFM handles temporal fusion on **registered** features.
>
> Thus, BRENet does not fuse misaligned modalities and rely on the network to implicitly correct misalignment; instead, it **corrects misalignment first and only then fuses**.
> While both approaches ultimately fuse RGB and event, fusion-based methods **assume** alignment and directly fuse features, whereas BRENet **enforces** alignment through explicit bidirectional registration *before* fusion. Our design treats the registration as the core objective where each module is proposed for or around registration instead of fusion. **We have revised descriptions of  Fig. 1(a) and the method section to make this distinction clearer.**

---

> ### Author Response · Authors · 2025-11-19
> **(2/2) Follow-up of Previous Comment**
>
> > RGB-based baselines and improvement
>
> We thank the reviewer for raising this important question. We clarify our choices and provide supporting evidence below:
>
> **(1) RGB-only baselines in Table 2 are not “outdated”**
>
> First, we respectfully note that, despite rapid progress in vision models, **there has not been a widely recognized, standardized, and lightweight RGB segmentation model after 2023 that is broadly adopted across robotics/event-based benchmarks**. Most recent works (e.g., SAM-based variants, Mask2Former-likes) are not directly suited to event datasets due to significant computational overhead.
>
> Second, according to **SE-Adapter (ICRA 2024)** which fine-tunes SAM for semantic segmentation under the same DSEC benchmark, the performance is:
>
> - **59.7 mIoU**,
> - **91.21% accuracy**.
>
> These are -**17.94 mIoU and -4.64% worse** than ours which is much worse than SegFormer. This confirms that **SegFormer remains a strong and competitive RGB baseline** for this setting, and newer large-scale models do not trivially surpass it for event-based segmentation.
>
> Third, we chose **SegFormer** because it **ensures fair comparison when using the *same backbone* compared to most of the existing works** without differences in architecture scale and depth. This aligns with the standard protocol in RGB-Event literature (CMX, CMNeXt, and EISNet).
>
> **(2) Scaling up RGB data is fundamentally limited**
>
> We appreciate the reviewer’s question about whether simply scaling up RGB data could surpass RGB-Event fusion. However, this assumption does not hold in practice because **RGB cameras significantly lose information due to sensor physics** (motion blur, low light, adverse weathers), which additional data cannot recover.
>
> Empirically:
>
> - **DSEC** (clear daytime): BRENet is **+4.7% mIoU** better than SegFormer.
> - **DELIVER** (adverse weather): BRENet achieves **63.13 mIoU**, **+10.4%** higher than SegFormer (57.20).
> - **DDD17** (low illumination): BRENet achieves **78.56 mIoU**, **+10.0%** higher than SegFormer (71.46).
>
> These large margins demonstrate that **event signals provide information that even 10× RGB data cannot recover**, particularly under challenging conditions. This is consistent with why event cameras are heavily used in drones, and autonomous systems.
>
> **(3) The improvements are not “marginal”**
>
> We respectfully disagree with the claim that improvements are marginal.
>
> In semantic segmentation:
>
> - **+1.0 mIoU** or **+0.3 accuracy** is typically considered a *significant* gain,
> - especially in saturated benchmarks.
>
> For example, the CVPR 2025 paper **SegMAN [1]**  reports improvements of only **+0.3 and+0.5 accuracy** on Cityscapes with MiT-B0/B2 backbones and these are considered meaningful advances.
>
> Under this context:
>
> - BRENet’s **+4.7 mIoU**, **+10.4 mIoU**, and **+10.0 mIoU** improvements
> - as well as **+0.53 and +0.73 accuracy gains**
>
> are **clearly substantial** and well beyond typical incremental improvements.
>
> > Results of RGB methods.
>
> We understand the reviewer's concern about fair comparison. The numbers are initially reported in EISNet and they claimed that they re-implemented RGB-only methods (SegFormer and SegNeXt) on the DDD17 and DSEC datasets and trained them from scratch. It enables fair comparison with our method.
>
>
> In summary, we have made these points clearer in the revised manuscript. We appreciate your valuable suggestions and hope that our responses have addressed your concerns and clarified the contributions of our work. We look forward to your responses soon.
>
> Best regards,
>
> Authors
>
> [1] SegMAN: Omni-scale Context Modeling with State Space Models and Local Attention for Semantic Segmentation

---

> ### Author Response · Authors · 2025-11-26
> **Looking Forward to Further Discussions with Reviewer ESLK**
>
> Thank you for your appreciation of our work. In the response, we have discussed our core motivation compared to fusion-centric methods in detail and demonstrated the baseline selections. We hope the revision paper has addressed your concerns and captured your interest. We look forward to engaging with you further.

---

### Official Review · Reviewer_ussM · 2025-10-29

**Soundness:** 3
**Presentation:** 3
**Contribution:** 3
**Rating:** 4
**Confidence:** 3

**Summary:**

This article aims to solve the semantic segmentation task of combining RGB images and event camera data. The author points out that the existing fusion-centric methods have inherent flaws as they ignore the inherent (i) Spatiotemporal Misalignment and (ii) Modal Misalignment issues between the two modalities. To address this issue, this article reconstructs the problem from "fusion" to "registration" and proposes a design principle of “registers first, then fuses.” Its core contribution is a novel flow-guided bidirectional registration framework called BRENet. The key component is the Motion-enhanced Event Tensor (MET) proposed by the author, which is a new event representation method that utilizes coarse-grained optical flow (as a guide) and fine-grained event temporal features to transform sparse event streams into dense representations. This framework uses MET representation in the Bidirectional Registration Module (BRM) and Temporal Fusion Module (TFM) to align and fuse features. Experiments on four large datasets (DDD17, DSEC, DELIVER, and M3ED) have shown that the proposed method achieves state-of-the-art performance.

**Strengths:**

1. The main advantage is the shift in the problem from the concept of "fusion" to "registration". This reconstruction directly solves a well-known fundamental challenge in RGB–Event perception (Spatiotemporal and Modal Misalignment), which many previous works have overlooked. The idea of converting inherent motion lag into optimizable optical flow estimation error is insightful.

2. The Motion-enhanced Event Tensor (MET) proposed by the author has clear motivation. It cleverly combines coarse-grained global motion (from optical flow) with fine-grained local temporal details (from Temporal Conv Module) to create a dense and easily registered representation. The ablation experiment (Table 5) confirmed its superiority over other representations such as voxel grid and AET.

3.  The proposed method demonstrated significant and consistent performance improvements compared to the previous SOTA model in four different and challenging benchmark tests.

4. The author provided a detailed ablation study that validated the contribution of each key component. The study clearly demonstrates the advantages of MET representation, the importance of coarse-grained optical flow and fine-grained temporal features, and the effectiveness of BRM and TFM. The plug-and-play verification of MET is also a bonus point.

**Weaknesses:**

1.  A major issue is that the framework relies on a pre-trained optical flow encoder (E-RAFT). The appendix mentions that the encoder was pre trained on the DSEC dataset. This raises a key question about comparative fairness, especially in DSEC benchmark testing. It is currently unclear to what extent the performance improvement is attributed to powerful, pre-trained prior knowledge of optical flow, rather than the BRENet architecture itself. The ablation experiments in Table 7 demonstrate robustness to different encoders, but they are all complex pre-trained models. The paper needs to clarify (a) how the performance would be if the optical flow encoder and segmentation task were trained together from scratch; (b) if the E-RAFT encoder has indeed been pre trained on DSEC, how does this not constitute data leakage for DSEC evaluation.
2.: The complete framework is very complex, involving multiple encoders (image, event temporal, optical flow), a CFE (Coarse-to-Fine Estimator), a bidirectional BRM, and a TFM. The use of bidirectional propagation and spatial warping increases the "significant computational cost" that cannot be ignored. Table 4 claims that MACs are "equivalent" to EISNet, but does not explicitly state whether the MACs of the optical flow encoder (E-RAFT) are included in this calculation. This negligence may be misleading. The paper needs to provide a more transparent delay analysis (such as FPS) and a clear breakdown of computational costs (including optical flow estimators).
3.  The design of the core modules (BRM and TFM) is an adaptation of existing work. The paper explicitly states that BRM is an extension of the Frequency-aware Cross-modal Feature Enhancement (FCFE) module in FEVD (Kim et al., 2024). TFM uses standard deformable convolutions for alignment. The main architectural innovation seems to be focused on the design of CFE/MET, rather than the registration and fusion modules themselves.
4. Failure case analysis (Section E) shows that in extremely dark areas, optical flow will "significantly degrade", leading to "blurry" predictions. This confirms that the performance of the model is closely related to the quality of the optical flow estimator. This is a direct and inherent weakness of the "flow guidance" method.

[1] Frequency-aware event-based video deblurring for real-world motion blur.CVPR2024

**Questions:**

Please seek weakness.

---

> ### Author Response · Authors · 2025-11-19
> **(1/2) Official Comment by Authors**
>
> We sincerely thank the reviewer for the thoughtful feedback and for recognizing the technical merit of our work. We appreciate that the reviewer thinks our paper is novel, has clear motivation, demonstrates significant improvements, and provides a detailed ablation study. Below we address each of the raised concerns in detail:
>
> > Pre-trained optical flow encoder (E-RAFT).
> >
>
> We thank the reviewer for raising this concern. We clarify that our improvements do not stem from a powerful flow prior, and this argument instead enhances our robustness:
>
> **(1) Improvements generalize beyond DSEC.**
>
> Although E-RAFT was originally trained on DSEC, the largest gains of BRENet occur **on *other* three datasets than DSEC**, where no such prior advantage exists:
>
> - **+3.53 mIoU on DDD17**,
> - **+4.63 mIoU on DELIVER**,
> - **+7.37 mIoU on M3ED**,
>
> demonstrating that BRENet’s benefits are **not tied to the DSEC domain** and generalize well across diverse RGB-Event benchmarks.
>
> **(2) No data leakage: E-RAFT was trained only on DSEC *training sequences*.**
>
> We emphasize that the public E-RAFT model is trained strictly on DSEC’s *training* split, which is disjoint from the DSEC *test* set used in our segmentation experiments. This is consistent with standard practice in event-based optical flow and does **not** constitute data leakage.
>
> **(3) BRENet’s gains are not due to a “powerful” flow network.**
>
> We deliberately chose E-RAFT **because it is *not* a large or state-of-the-art flow estimation model. I**t is a lightweight design from 2021 with only **5M parameters**, trained on **7,000 images**. As shown in **Table 8** from the paper, replacing E-RAFT with alternative flow encoders changes performance by only **±0.42 mIoU**, which is **20% of BRENet’s total improvement**. This indicates that the flow prior provides *only* coarse motion cues, while the majority of the gain arises from the BRENet architecture, particularly the registration-centric design that converts flow into an effective cross-modal alignment mechanism.
>
> **(4) End-to-end training confirms the result.**
>
> We additionally tested **end-to-end training** without using a pre-trained E-RAFT in Table 1 below. The performance is comparable relative to using pre-trained E-RAFT, further demonstrating that the pre-trained flow network is *not* the primary source of improvement. BRENet continues to outperform fusion-based baselines (73.07 mIoU of EISNet) even when the flow encoder is trained from scratch.
>
> Table 1: Ablation study under end-to-end training setting
>
> | Setting | mIoU $\uparrow $ | Gain vs SOTA (EISNet) |
> | --- | --- | --- |
> | E2E | 74.62 | (+1.55) |
> | Pretraining | 74.94 | (+1.87) |
>
> > Delay analysis and parameter size.
> >
>
> Thanks for the insightful suggestion. We agree that the inference speed is significant and we conduct new experiments for latency across several SOTA models below in Table 2. Compared to SOTA methods, our model achieves comparable latency and only slightly higher than the most efficient model while providing significantly lower GFLOPs and higher performance in mIoU, demonstrating that the additional registration components introduce *minimal* overhead in practice.
>
> Table 2: Model complexity on DDD17 dataset
>
> |  | Latency (ms)$\downarrow $ | Params (M)$\downarrow $ | GFLOPs $\downarrow $ | mIoU$\uparrow $ |
> | --- | --- | --- | --- | --- |
> | CMX | 104.2 | 66.56 | 64.9 | 67.47 |
> | CMNeXt | 101.7 | 58.68 | 65.3 | 66.99 |
> | EISNet  | 83.5 | 34.39 | 69.2 | 75.03 |
> | Ours | 94.3  | 37.69 | 55.2 | 78.56 |
>
> To further analyze each component, we conduct a detailed parameter, GFLOPs and their ratios analysis below in Table 3. It further demonstrates our design’s efficiency without introducing heavy modules. Our proposed core contribution of using Flow encoder and CFE only consists of 16.8% parameter size and 28.3% GFLOPs.
>
> Table 3: Detailed model complexity analysis on DDD17
>
> |  | Params (M) $\downarrow $ | Params Ratio | GFLOPs | GFLOPs Ratio |
> | --- | --- | --- | --- | --- |
> | Image backbone | 24.2 | 64.2% | 4.5 | 8.2% |
> | Flow encoder | 5.3 | 14.1% | 10.1 | 18.3% |
> | CFE | 1.0 | 2.7% | 5.5 | 10.0% |
> | BRM | 2.6 | 6.9% | 20.7 | 37.5% |
> | TFM | 1.6 | 4.2% | 7.1 | 12.9% |
> | Other layers | 3.0 | 7.9% | 7.3 | 13.2% |

---

> ### Author Response · Authors · 2025-11-19
> **(2/2) Follow-up of Previous Comment**
>
> > Design of the BRM and TFM modules.
>
> We thank the reviewer for the thoughtful comments on the design of BRM and TFM. While these modules are indeed inspired by prior work (e.g., FCFE and deformable convolutions), our main novelty lies at the *conceptual* level and MET rather than these two blocks. Specifically, our contributions are two-fold:
>
> 1. **Paradigm shift from fusion-centric to registration-centric RGB-Event perception.**
>
>     Prior RGB–Event methods largely treat events as an additional modality to be fused with RGB features under an implicit alignment assumption. In contrast, we recast the task as a **registration problem**, and design the entire pipeline around a *register → fuse* principle. BRM and TFM are not generic enhancement blocks: they are explicitly constructed to realize this registration-centric view and can be replaced by any other fusion blocks.
>
> 2. **A new event representation (MET) and a redefined role of optical flow.**
>
>     We introduce **Motion-Enhanced Event Tensors (MET)** and use optical flow **not as an extra input modality**, but as a **bridge** that dynamically aligns events with RGB in space and time. To the best of our knowledge, we are the first to employ optical flows explicitly for RGB-Event perception tasks. MET, BRM, and TFM are jointly designed around this idea of flow-guided registration.
>
>
> We have revised the introduction and method sections to emphasize that the key innovation is this **registration-centric formulation and flow-as-bridge design**, with BRM/TFM serving as concrete mechanisms to implement it, rather than minor adaptations of existing fusion modules.
>
> > Extremely dark failure cases.
> >
>
> We thank the reviewer for highlighting the failure cases in Section E. We agree that optical flow may degrade in *extremely* dark regions. However, we would like to clarify several points that mitigate the concern that BRENet is “mostly dependent” on flow quality:
>
> **(1) BRENet remains robust in modest and even fairly dark conditions.**
>
> Under modest low-light conditions, BRENet maintains strong segmentation accuracy even
> when flow estimation becomes noisy, showing that the flow-guided registration is effective under realistic low-light conditions. Moreover, in the *extreme* low-light examples shown in **Figure 13**, most regions are still segmented reasonably well and are comparable to baselines; only a few localized areas with almost no signal exhibit the “blurry” predictions. This indicates that the model is not uniformly failing whenever illumination is low.
>
> **(2) The failures correspond to extreme illumination collapse, where all modalities struggle.**
>
> The failure cases in Section E occur when both RGB and event sensors receive **near-zero signal**, making segmentation difficult for any architecture or modality. In such scenarios, it is not accurate to conclude that performance is dominated by flow quality alone: the underlying issue is that **there is barely any visual/event information to recover**, so both flow and semantic prediction are ill-posed. Thus, these examples highlight a limitation of sensing under extreme conditions rather than a specific vulnerability of the flow-guided design.
>
> **(3) Focusing solely on extreme darkness is out of our task scope.**
>
> Our work targets **RGB–Event semantic segmentation**, not extreme low-light or nighttime-only segmentation. While we intentionally report failure cases for transparency, evaluating the method primarily in such extreme scenarios is not aligned with our scope. On standard benchmarks that contain a mixture of normal, moderately low-light scenes, and multiple weather conditions (DDD17, DELIVER, M3ED), BRENet still achieves **+3.53, +4.63, and +7.37 mIoU** gains over strong baselines, demonstrating robust and practical benefits in typical RGB-Event conditions.
>
> In summary, although optical flow naturally degrades when sensors receive almost no signal, BRENet is robust in modest and even fairly dark environments, and the extreme failures arise from sensor-level limitations rather than an inherent weakness of flow-guided registration.
>
> Once again, thank the reviewer for these constructive feedback. The comments have significantly improved the clarity and impact of our work. We hope our responses adequately address all concerns and look forward to further responses.
>
> Best regards,
>
> Authors

---

> ### Author Response · Authors · 2025-11-26
> **Looking Forward to Further Discussions with Reviewer ussM**
>
> Thank you for your time and attention. In the response, we added additional runtime analysis and ablation studies of end-to-end training. We have also clarified and addressed several misunderstandings like data leakage and extremely dark failure cases. We hope you have had the opportunity to review our revisions and look forward to further engaging discussions with you.

---

### Official Review · Reviewer_Z6i3 · 2025-10-30

**Soundness:** 3
**Presentation:** 3
**Contribution:** 2
**Rating:** 6
**Confidence:** 3

**Summary:**

This paper reframes RGB–event segmentation from fusion to registration: estimate forward/backward optical flow on the event stream, build a Motion-enhanced Event Tensor (MET) that mixes coarse (flow) and fine (temporal event) cues, register events to the RGB frame in both directions, then fuse temporally (TFM). Results on DDD17, DSEC, DELIVER, and M3ED are consistently strong, with SOTA performances on all four datasets.

**Strengths:**

1. The idea of leveraging optical flow to help event-frame registration is interesting.
2. The performances of the method on the frame-event datasets are very compelling.
3. The motivation of the method is clear.

**Weaknesses:**

Major:
1. Backward “flow” is obtained by reversing the order of the event frames to compute $O^b$. That means inference uses future events inside each window, which limits true online deployment and complicates latency guarantees.
2. The authors claim “comparable processing latency” but do not report FPS or wall-clock latency, only Params/MACs. This is not convincing enough.
3. It's not clear whether the rise of feature similarity in Tab. 1 results from the optical flow itself. It could also be attributed to the dense property of optical flow compared to the voxel grid.

Minor:

The contraction argument around the flow refinement (Eq. 5) is heuristic and not tied to specific network conditions. There’s no theorem that bounds segmentation error as a function of measurable flow error on these datasets.

**Questions:**

Please refer to Weaknesses

---

> ### Author Response · Authors · 2025-11-19
> **Official Comment by Authors**
>
> We sincerely thank the reviewer for the thoughtful and constructive comments. The feedback helped us significantly strengthen the paper. In the revised version, we clarified the design decisions behind our registration-centric framework, added new complexity analysis, and refined our explanations on feature similarity:
>
> > **Clarification on backward flow and online deployment.**
> >
>
> We thank the reviewer for raising this point. We would like to clarify that the *backward* flow is obtained by reversing the order of the **past event frames** and both the forward and backward alignments are computed using **only observed history**. This does *not* involve future information beyond the current timestamp and therefore **does not cause data leakage**, nor does it prevent online deployment. As discussed in Section 3.3, our registration-centric design supports **online inference**, and in practice the system shows strong accuracy with **comparable latency**.
>
> > **Latency measurement.**
> >
>
> We appreciate the reviewer’s thoughtful suggestion regarding runtime. Following this feedback, we conducted new experiments to measure **inference latency** (ms). The results are reported in **Table 1** below. Compared to SOTA methods, our model achieves comparable latency and only slightly higher than the most efficient model while providing significantly higher performance in mIoU, demonstrating that the additional registration components introduce *minimal* overhead in practice.
>
> Table 1: Model complexity on DDD17 dataset
>
> |  | Latency (ms)$\downarrow $ | mIoU $\uparrow $ |
> | --- | --- | --- |
> | CMX | 104.2 | 67.47 |
> | CMNeXt | 101.7 | 66.99 |
> | EISNet  | 83.5 | 75.03 |
> | Ours | 94.3 | 78.56 |
>
> > **Feature similarity analysis.**
> >
>
> We thank the reviewer for the comment. This statement is aligned with our thoughts and observations.
>
> The increased similarity is not only from the density of flows but also from the **unique role flow plays in our framework**, specifically:
>
> - **(i) Flow converts sparse, discrete event features into dense, continuous motion-aligned representations**, making them compatible with dense, continuous RGB features and reducing modal misalignment.
> - **(ii) Flow enables temporal registration:** it warps event pixels from a sequence of past timestamps into the current RGB frame, reducing spatiotemporal discrepancies by registration.
> - **(iii) Flow shifts irreducible RGB-Event misalignment into a form of *flow estimation errors***, which can be optimized by our model.
>
> These characteristics directly support the observed rise in feature similarity after registration. We have clarified this motivation and its empirical evidence more explicitly in the revised manuscript.
>
> We appreciate the reviewer’s insights, which align closely with our motivations and observations, and we believe the additional clarifications and experiments directly address the concerns raised. Look forward to the further responses.
>
> Sincerely,
>
> Authors

---

> > ### Comment · Reviewer_Z6i3 · 2025-11-25
> >
> > Thanks for your responses. Most of my concerns have been addressed, but I still have a question regarding Feature similarity analysis: Are there any experiments proving these points? (e.g., comparing optical flow to some dense representations like frame images)

---

> ### Author Response · Authors · 2025-11-26
> **Follow-up Response to Reviewer Z6i3’s Question on Feature Similarity Analysis**
>
> Thanks for the insightful suggestion. We conducted additional feature similarity analysis between optical flow and  6-Channel Image in Table 1 below. It demonstrates that compared to voxel grids, dense representations like frame images reduce the sparse-dense appearance misalignment, but it still has gap and these pseudo-dense representations do not provide truly continuous visual information like optical flow. This is aligned with our observation and we have updated the table 1 in the paper.
>
> Table 1: Comparison of feature similarity on DDD17 and DSEC. Higher values
> indicate stronger cross-modal alignment.
> |  | DDD17 | DSEC |
> | --- | --- | --- |
> | Voxel grid - RGB | 0.037 | 0.009 |
> | 6-Channel Image - RGB | 0.053 | 0.016 |
> | Optical flow - RGB | 0.127 | 0.071 |

---

### Official Review · Reviewer_fLRQ · 2025-10-31

**Soundness:** 2
**Presentation:** 3
**Contribution:** 2
**Rating:** 4
**Confidence:** 4

**Summary:**

This paper introduces BRENet, a flow-guided, registration-centric framework for RGB–Event semantic segmentation. Unlike previous fusion-based methods that assume perfect alignment between RGB and event data, BRENet explicitly addresses spatiotemporal and modal misalignment by reformulating the task as a registration problem. The model uses optical flow as a bridge between modalities and proposes three key components: Motion-Enhanced Event Tensor (MET) for dense temporal representation, Bidirectional Registration Module (BRM) for forward/backward alignment, and Temporal Fusion Module (TFM) for temporally coherent fusion. Experiments on four benchmarks show consistent improvements over state-of-the-art baselines.

**Strengths:**

(1) Clear motivation and paradigm shift. The paper provides a well-articulated motivation for redefining RGB–Event perception from a fusion problem to a registration problem. The analysis of spatiotemporal and modal misalignment is well grounded, and the use of optical flow as a registration bridge is novel and conceptually elegant.

(2) Comprehensive and well-validated framework. BRENet integrates several complementary modules (MET, BRM, TFM) in a coherent design. The experimental validation is extensive, covering multiple benchmarks and ablation studies.

**Weaknesses:**

(1) Model complexity and runtime considerations. BRENet involves multiple heavy components—bidirectional flow estimation, frequency-domain operations, and deformable convolutions—making the pipeline complex. Although parameter counts and MACs are reported, there is no analysis of inference speed or runtime efficiency, which is critical for real-time event-based perception. It is necessary to provide relevant data to evaluate the complexity of the model.

(2) Lack of analysis experiments on architectural or hyperparameter settings. Although the model includes several tunable components, the paper lacks sensitivity or analysis studies. For instance, it would strengthen the work to analyze: Bin size (B) and its impact on event density and segmentation quality; Number of refinement iterations (J) in the Coarse-to-Fine Estimator; Frequency-domain fusion settings (e.g., FFT resolution or real/imaginary channel ratios). Such experiments would clarify the model’s stability and provide deeper insights into why each module design choice matters.

(3) The limitation section in the appendix is brief and does not thoroughly discuss the dependence on pre-trained flow encoders, potential propagation of flow errors, or scalability to other modalities (e.g., depth, LiDAR).

**Questions:**

See the weakness.

---

> ### Author Response · Authors · 2025-11-19
> **(1/2) Official Comment by Authors**
>
> We sincerely thank the reviewer for the thoughtful and detailed feedback. We are encouraged by the positive remarks regarding our motivation, methodology, and comprehensive results. We are committed to making improvements in the final version and providing any additional experiments or analyses that strengthen our claims. We provide clarifications and evidence that address each of the raised concerns in detail below:
>
> > Model complexity and runtime considerations.
>
> Thanks for the insightful suggestion. We agree that the inference speed is significant and we conduct new experiments for inference latency across several SOTA models below in Table 1. Compared to SOTA methods, our model achieves comparable latency and only slightly higher than the most efficient model while significantly outperforming them in mIoU.
>
> Table 1: Model complexity on DDD17 dataset
>
> |  | Latency (ms) $\downarrow $ | mIoU $\uparrow $ |
> | --- | --- | --- |
> | CMX | 104.2 | 67.47 |
> | CMNeXt | 101.7 | 66.99 |
> | EISNet  | 83.5 | 75.03 |
> | Ours | 94.3 | 78.56 |
>
> > Lack of analysis experiments on architectural or hyperparameter settings.
> >
>
> We thank the reviewer for the insightful suggestions regarding additional ablation studies. In response, we have either conducted the corresponding experiments or provided further clarifications where appropriate.
>
> **(1) Bin size (B).** We apologize if this was not sufficiently highlighted: our ablation on bin size B is already included in the supplementary material (Table 9). We evaluate B={1,3,5,15} and observe stable performance with small variance across this range, demonstrating that MET is not sensitive to the choice of temporal bin size. We have moved this analysis into the main paper for better visibility.
>
> **(2) Refinement iterations (J).** We appreciate the reviewer’s close reading and we want to clarify this misunderstanding. The notation **J** appears in Section 3.1 as part of the *standard training formulation* used for theoretical motivation. Importantly, **J** is **conceptual for training deep learning models** and does **not** correspond to any iterative refinement module or tunable hyperparameter in BRENet. BRENet does not perform iterative flow updates. We have revised Section 3.1 to explicitly clarify that **J** is conceptual and not part of our architecture.
>
> **(3) Frequency-domain settings.** Following the reviewer’s suggestion, we conducted new experiments analyzing real/imaginary channel ratios of CFE in Table 2 below:
>
> Table 2: Ablation study of real and imaginary channel ratios on DSEC
>
> |  | mIoU $\uparrow $ |
> | --- | --- |
> | Real-only | 74.31 |
> | Imaginary-only | 73.96 |
> | Ours | 74.94 |
>
> Specifically, we compared **real-only**, **imaginary-only**, and **real+imaginary** configurations. It demonstrates that both parts are important for fusion on the frequency domain and the full real+imaginary combination achieves the strongest results. Note that even the imaginary-only option achieves superior performance than SOTA (73.07 mIoU of EISNet) by +0.89 mIoU.
>
> We believe these analyses directly address the reviewer’s concern and further demonstrate the architectural stability and robustness of BRENet.

---

> ### Author Response · Authors · 2025-11-19
> **(2/2) Follow-up of Previous Comment**
>
> > Discussion of the dependence on pre-trained flow encoders and potential propagation of flow errors.
>
> We appreciate this important question. To further explain the robustness of our design, we make new clarifications in “Selection of different flow encoders” of Section  4.5 in our newly uploaded paper based on our previous ablation studies of that in supplementary materials. In summary, our method is designed to **tolerate imperfect flows** by unique functionality over flow precision. The consistent improvements prove that even **approximate motion cues of flows enhance RGB-Event registration**. We additionally provide **challenging cases** in low-light conditions.
>
> First, our framework adopts E-RAFT, which is non-SOTA from 2021, to avoid leveraging the latest advancements. This design choice was intentional—to demonstrate the **robustness and generalizability** of our approach without relying on cutting-edge flow quality. As shown in Table 3 below, optical flows are not utilized for pixel-level precision of feature aggregation but rather as a **bridge and coarse visual guidance for registration.** Specifically, it provides flow-aware cues to temporally align asynchronous event streams and convert sparse events into **dense, motion-enhanced representations**. As shown in Tables 5 & 10 from the paper, introducing optical flows into the RGB-Event baseline improves segmentation performance even when it is imprecise, as long as the optical flows are not entirely erroneous. Furthermore, **Table 3 below** demonstrates that MET maintains stable performance across various flow backbones, highlighting that our model does not rely on high-quality optical flow, but instead benefits from the **inherent structural registration**.
>
> Table 3: Ablation study of different optical flow encoders
>
> |  | Publication | mIoU $\uparrow $ |
> | --- | --- | --- |
> | **BRENet + E-RAFT** | 3DV 2021 | 74.94 |
> | **BRENet + TMA [6]** | ICCV 2023 | 74.62 |
> | **BRENet + ADMFlow [7]** | ICCV 2023 | 75.11 |
> | **BRENet + EEMFlow [8]** | CVPR 2024 | 75.36 |
>
> Additionally, we perform **zero-shot evaluations on DSEC-Night** for challenging low-light scenarios in Section E. These experiments reveal that our model maintains strong segmentation accuracy even when flow estimation becomes noisy as nighttime. Specifically, in **moderately dark regions**, **events remain robust** and support **reliable flow estimation** to guide RGB features.
>
> In conclusion, our model is robust to the potential flow errors due to its role in our unique design: as a bridge that convert the inherent, irreducible spatial shift to estimation errors and dynamically aligns event with RGB rather than an alternative input modality.
>
> Thank you again for your constructive feedback. Your suggestions and comments have been insightful in refining the clarity and quality. We hope our responses address your concerns comprehensively and look forward to your further response.
>
> Sincerely,
>
> Authors

---

> > ### Comment · Reviewer_fLRQ · 2025-11-26
> >
> > Thanks for the detailed response. While you provided latency comparisons, this does not fully address the concern about the architectural complexity of BRENet. Besides, the discussion on robustness to flow errors is helpful, but still incomplete. The key concern was not only backbone choice, but the error propagation mechanism. Looking forward to the detailed discussion and analysis.

---

> ### Author Response · Authors · 2025-11-26
> **Looking Forward to Further Discussions with Reviewer fLRQ**
>
> Thank you for your valuable suggestions. In the response, we have incorporated additional runtime analysis and ablation studies. We have also improved the clarifications of the dependence on pre-trained flow encoders. We hope you have had the chance to review these updates and look forward to engaging in further discussions with you.

---

> ### Author Response · Authors · 2025-11-27
> **Follow-up Response to Reviewer fLRQ’s Question on Complexity and Flow Error**
>
> We sincerely thank the reviewer for the follow-up question. We provide further clarifications below.
>
> > Complexity
>
> We further conduct a detailed parameter, GFLOPs and their ratios analysis below in Table 1. Our proposed core contribution of using **Flow encoder** and **CFE** only consists of **14.1%** and **2.7%** parameter size and **18.3%** and **10.0%** GFLOPs. It demonstrates our core contribution’s efficiency without introducing heavy modules.
>
> Table 1: Detailed model complexity analysis on DDD17
>
> |  | Params (M) $\downarrow $ | Params Ratio | GFLOPs $\downarrow $ | GFLOPs Ratio |
> | --- | --- | --- | --- | --- |
> | Image backbone | 24.2 | 64.2% | 4.5 | 8.2% |
> | Flow encoder | 5.3 | 14.1% | 10.1 | 18.3% |
> | CFE | 1.0 | 2.7% | 5.5 | 10.0% |
> | BRM | 2.6 | 6.9% | 20.7 | 37.5% |
> | TFM | 1.6 | 4.2% | 7.1 | 12.9% |
> | Other layers | 3.0 | 7.9% | 7.3 | 13.2% |
>
> > Flow Error
>
> (1) We provide a rigorous proof using the **Lipschitz Continuity**, demonstrating that the final segmentation error is linearly bounded by flow errors.
>
> Let the CFE module be defined as a function $\Phi_{CFE}$ that maps the estimated optical flow $U$ to the Motion-Enhanced Event Tensor $M$ then to final output $Y$. Since CFE consists of standard differentiable layers (MLP, DeformConv), it is Lipschitz continuous with constant $K_{CFE}$.
>
> $Y = \Phi_{fusion}(\Phi_{CFE}(U))$
>
> $\| M - \hat M \| \le K_{CFE} \cdot \| U -\hat U \|$
>
> Due to the smoothness of feature maps, $K_{CFE} << 1$. Similarly, in the following fusion modules (BRM + TFM), we define
>
> $Y = \Phi_{fusion}(M) \implies \| Y - \hat Y \| \le K_{fusion} \cdot \| M - \hat M\|$
>
> To find the relationship between the flow errors $\epsilon$ and the final output error $\Delta Y$, we then substitute the bound from above, therefore the Total Error Bound is:
>
> $\| Y - \hat Y\| \le K_{fusion} \cdot (K_{CFE} \cdot \| U - \hat U \|)$
>
> $\| \Delta Y \| \le (K_{fusion} \cdot K_{CFE}) \cdot \epsilon$
>
> In conclusion, the model is **Lipschitz Stable**. The error propagation is linearly bounded and stable, governed by the constant factor $K_{total} = K_{fusion} \cdot K_{CFE}$. The final error will not explode neither.
>
> (2) To further verify the impact of error propagation, we intentionally insert Gaussian Noise (0.1, 0.2, and 0.5 standard deviation) into each pixel of flow maps during inference following the common practice [1]. The results are shown in Table 2 below. Even the highest-level noise will only reduce performance by 0.71 mIoU which is still 1.16 higher than the previous best. It demonstrates that our model is robust to flow errors and our design with self-correction will not propagate the significant errors.
>
> Table 2: Ablation study of adding Gaussian Noise to flow maps
>
> | Model | Std| DSEC |
> | --- | --- | --- |
> | BRENet | - | 74.94 |
> | BRENet | 0.1 | 74.73 |
> | BRENet | 0.2 | 74.55 |
> | BRENet | 0.5 | 74.23 |
>
> (3) Ablation studies of different flow encoders demonstrates alternative flow encoders changes performance by only **±0.42 mIoU**, which is **20% of BRENet’s total improvement**. This indicates that the flow prior provides *only* coarse motion cues, and potential flow errors will not significantly impact the final accuracy under our **inherent structural registration**.
> (4) We additionally tested **end-to-end training** without using a pre-trained E-RAFT in Table 3. It demonstrates that the pre-trained flow network is *not* the primary source of improvement. BRENet continues to outperform fusion-based baselines (73.07 mIoU of EISNet) even when the flow encoder is trained from scratch.
>
> Table 3: Ablation study under end-to-end training setting
>
> | Setting | mIoU $\uparrow $ | Gain vs SOTA (EISNet) |
> | --- | --- | --- |
> | E2E | 74.62 | (+1.55) |
> | Pretraining | 74.94 | (+1.87) |
>
> [1] Benchmarking Multi-modal Semantic Segmentation under Sensor Failures: Missing and Noisy Modality Robustness

---

### Author Response · Authors · 2025-11-24
**Summary and Answers of Official Reviews (1/2)**

Dear Reviewers,

We sincerely thank all reviewers for their constructive comments and insights. We appreciate the recognition of our paper and the valuable suggestions for enhancing the quality. Below, we summarize the key strengths of our paper by reviewers in Table 1. These acknowledgements motivate us to further improve the quality and impact of our work.

Table 1: Recognized Contributions.

| **Contribution** | **Reviewer** | **Official Review** |
| --- | --- | --- |
| **1. Clear Motivation** | fLRQ | "The paper provides a well-articulated **motivation**…" |
|  | Z6i3 | "The **motivation** of the method is clear." |
|  | ussM | "The Motion-enhanced Event Tensor (MET) proposed by the author has clear **motivation**." |
| **2. Paradigm Shift** | fLRQ | "Clear motivation and **paradigm shift**." |
|  | Z6i3 | "This paper **reframes** RGB-event segmentation from fusion to registration…" |
|  | ussM | "The main advantage is the **shift in the problem** from the concept of "fusion" to "registration"." |
|  | ESLK | "…where the proposed one finds **better pixel-level correspondences**" |
| **3. Novel and Interesting Idea** | fLRQ | "…the use of optical flow as a registration bridge is **novel and conceptually elegant**." |
|  | Z6i3 | "The idea of leveraging optical flow to help event-frame registration is **interesting**." |
|  | ussM | "The idea of converting inherent motion lag into optimizable optical flow estimation error is **insightful**." |
|  | ESLK | "The idea of using optical flow to align event data with RGB data is **interesting**." |
| **4. Strong Performance Improvement** | fLRQ | "Experiments on four benchmarks show **consistent improvements** over state-of-the-art baselines." |
|  | Z6i3 | “The performances of the method on the frame-event datasets are very **compelling**.” |
|  | ussM | “The proposed method demonstrated **significant and consistent performance** improvements…” |
|  | ESLK | “…leading to better representation and **performance improvement**...” |
| **5. Thorough Analysis** | fLRQ | “The experimental validation is **extensive**, covering multiple benchmarks and ablation studies.” |
|  | ussM | “The author provided a **detailed** ablation study…” |

Thank you again for your support and suggestions, which have greatly improved the quality of our paper.

Best regards,

Authors

---

### Author Response · Authors · 2025-11-24
**Summary and Answers of Official Reviews (2/2)**

Thank all the reviewers for the insightful questions and careful review which help us to improve the quality of our work. Here we answer the questions of common concern.

### **1. Runtime considerations and Practical Deployability**

We conduct new experiments for latency across several SOTA models below in Table 1. Compared to SOTA methods, our model achieves comparable latency and only slightly higher than the most efficient model while significantly outperforming them in mIoU.

Table 1: Model complexity on DDD17 dataset

|  | Latency (ms) $\downarrow $ | mIoU $\uparrow $ |
| --- | --- | --- |
| CMX | 104.2 | 67.47 |
| CMNeXt | 101.7 | 66.99 |
| EISNet  | 83.5 | 75.03 |
| Ours | 94.3 | 78.56 |

in addition, we would like to clarify that the *backward* flow is obtained by reversing the order of the **past event frames** and they are computed using **only observed history**. This does *not* involve future information beyond the current timestamp and therefore **does not cause data leakage**, nor does it prevent online deployment. As discussed in Sec. 4.4, our registration-centric design supports **online inference**, and in practice the system shows strong accuracy with **comparable latency**.

### **2. Dependence on pre-trained flow encoders**

Our method is designed to **tolerate imperfect flows.**

First, our framework adopts E-RAFT, which is non-SOTA from 2021, to avoid leveraging the latest advancements. It was pre-trained on 7k images with 5M parameter which is not powerful at all. It demonstrates the **robustness and generalizability** of our approach without relying on cutting-edge flow quality. Furthermore, **Table 2 below** demonstrates that MET maintains stable performance across various flow backbones and replacing E-RAFT with alternative flow encoders changes performance by only **±0.42 mIoU**, which is **20% of BRENet’s total improvement**. This indicates that the flow prior provides *only* coarse motion cues, while the majority of the gain arises from the **inherent structural registration**.

Table 2: Ablation study of different optical flow encoders

|  | Publication | mIoU $\uparrow $ |
| --- | --- | --- |
| **BRENet + E-RAFT** | 3DV 2021 | 74.94 |
| **BRENet + TMA [6]** | ICCV 2023 | 74.62 |
| **BRENet + ADMFlow [7]** | ICCV 2023 | 75.11 |
| **BRENet + EEMFlow [8]** | CVPR 2024 | 75.36 |

We additionally tested **end-to-end training** without using a pre-trained E-RAFT in Table 3. The performance is comparable relative to using pre-trained E-RAFT, further demonstrating that the pre-trained flow network is *not* the primary source of improvement. BRENet continues to outperform fusion-based baselines (73.07 mIoU of EISNet) even when the flow encoder is trained from scratch.

Table 3: Ablation study under end-to-end training setting

| Setting | mIoU $\uparrow $ | Gain vs SOTA (EISNet) |
| --- | --- | --- |
| E2E | 74.62 | (+1.55) |
| Pretraining | 74.94 | (+1.87) |

---

### Meta-Review · Area_Chair_bVBN · 2025-12-13

**Summary:**

I regret to recommend rejection of this submission. The paper proposes a flow-guided, registration-first framework (BRENet) for RGB–Event semantic segmentation, reframing prior fusion-centric pipelines as explicit cross-modal registration and introducing MET together with bidirectional registration and temporal fusion modules. Reviewers generally recognize the motivation and the empirical improvements. However, the current record still leaves substantial uncertainty around (i) overall system complexity and deployability (beyond reporting latency), (ii) the precise online/causal interpretation and evaluation protocol for the “backward” direction within an event window, and (iii) robustness to flow quality and error propagation, i.e., how tightly the gains depend on the flow component/backbone under realistic noise and failure cases. Finally, after the system bug occurred, authors and reviewers did not have a further opportunity to continue discussion, so I must base the decision on the existing evidence and unresolved concerns.

**Reviewer Concerns:**

Concerns that were addressed (at least partially) in the rebuttal/discussion:

- Efficiency evidence: the authors added wall-clock latency comparisons and additional efficiency-related results.

- Causality clarification: the authors clarified that the “backward” direction is computed by reversing past events (no future events beyond the current timestamp).

- Attribution tests: the rebuttal added comparisons intended to support that the observed alignment/similarity gains stem from flow-guided registration rather than simply using denser representations.

- Robustness (partial): additional analysis and noise-injection style experiments were provided to suggest limited degradation under flow noise.


Concerns that remain outstanding (and are decision-critical):

- System complexity / deployability: even with added latency numbers, concerns persist about the complexity of a multi-module pipeline, interaction effects between components, and maintainability for real deployment.

- Online/streaming protocol: while “no future events” is clarified, the paper would benefit from a clearly specified streaming evaluation protocol (buffering assumptions, strict causality, and end-to-end online latency/accuracy) to remove remaining ambiguity.

- Flow dependence and error propagation: the added evidence is helpful, but it is not yet fully conclusive that gains are stable under realistic flow errors and failure modes, nor that performance is not tightly coupled to the flow backbone/pretraining.

- Incomplete visibility into one review’s detailed weaknesses (truncated text) adds uncertainty rather than confidence in closure.

**Reviewer Scores:**

fLRQ: 4 → 4 (at most 4 → 6). The rebuttal mitigates some points (latency/robustness), but the core “complex system” concern likely remains.

Z6i3: 6 → 6. Clarifications help, but a clear score increase would likely require an explicit streaming/online evaluation protocol.

ussM: 4 → 4 (at most 4 → 6). Added experiments may improve confidence slightly, but concerns around complexity and flow-dependence likely persist.

ESLK: 4 → 4 (at most 4 → 6). The best estimate is no change.

---

### Decision · Program_Chairs · 2026-01-26

Reject